# Virtual Scanning: Unsupervised Non-line-of-sight Imaging from Irregularly Undersampled Transients

**Xingyu Cui**[1]    **Huanjing Yue**[1]    **Song Li**[2,3]    **Xiangjun Yin**[1]
**Yusen Hou**[1]    **Yun Meng**[2,3]    **Kai Zou**[2,3]    **Xiaolong Hu**[2,3]    **Jingyu Yang**[1,*]

[1]School of Electrical and Information Engineering, Tianjin University, China
[2]School of Precision Instrument and Optoelectronic Engineering, Tianjin University, China
[3]Key Lab. of Optoelectronic Information Science and Technology, Ministry of Education, China

## Abstract

Non-line-of-sight (NLOS) imaging allows for seeing hidden scenes around corners through active sensing. Most previous algorithms for NLOS reconstruction require dense transients acquired through regular scans over a large relay surface, which limits their applicability in realistic scenarios with irregular relay surfaces. In this paper, we propose an unsupervised learning-based framework for NLOS imaging from irregularly undersampled transients (IUT). Our method learns implicit priors from noisy irregularly undersampled transients without requiring paired data, which is difficult and expensive to acquire and align. To overcome the ambiguity of the measurement consistency constraint in inferring the albedo volume, we design a virtual scanning process that enables the network to learn within both range space and null space for high-quality reconstruction. We devise a physics-guided SURE-based denoiser to enhance robustness to ubiquitous noise in low-photon imaging conditions. Extensive experiments on both simulated and real-world data validate the performance and generalization of our method. Compared with the state-of-the-art (SOTA) method, our method achieves higher fidelity, greater robustness, and remarkably faster inference times by orders of magnitude. The code and model are available at https://github.com/XingyuCuii/Virtual-Scanning-NLOS.

## 1 Introduction

Non-line-of-sight (NLOS) imaging aims to reconstruct hidden scenes beyond the direct line of sight of the detector, garnering interest across various fields such as robot vision, autonomous driving, rescue operations, remote sensing, and medical imaging [1–5]. In a typical active confocal NLOS imaging system, as depicted in Fig. 1, a laser source and a detector are both focused on the same point on a relay surface. Pulses emitted by the laser reflect off the surface to illuminate the hidden scene. The detector captures photons bouncing back from the scene toward the relay surface, referred to as transients, from which the hidden scene can be recovered using elaborately designed algorithms.

While existing works have achieved remarkable breakthroughs, they also face significant limitations that hinder their practical applicability. These methods assume dense and regular scanning of a large relay surface, which may not be feasible in realistic scenarios with irregular relay surfaces such as latticed windows or fences. Transients obtained through irregular undersampling can result in severe ill-posedness, leading to artifacts or reconstruction failure. This raises the challenging task of *NLOS imaging from irregularly undersampled transients (IUT)*. To address this, Liu et al. [6] proposed introducing manually designed strong regularization terms under a functional optimization framework. However, this approach is hindered by the need for lengthy numerical iterative computations. Recent

---

*Corresponding author: yjy@tju.edu.cn.

38th Conference on Neural Information Processing Systems (NeurIPS 2024).

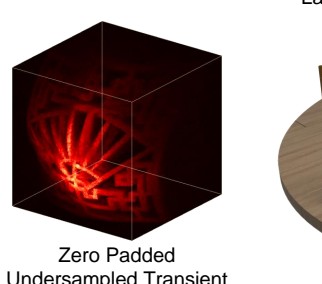
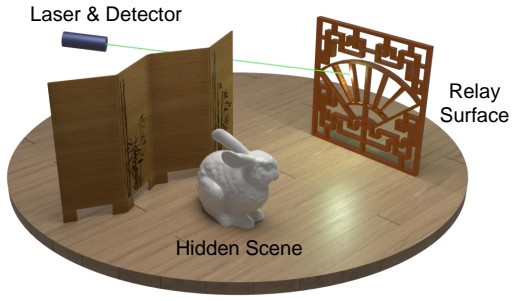
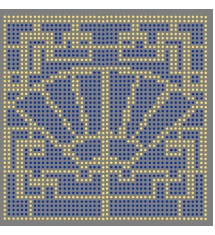

Laser & Detector

Relay Surface

Hidden Scene

Zero Padded Undersampled Transient

Scannable Points

Figure 1: Illustration of the active confocal NLOS imaging with an irregular relay surface. Yellow points indicate scannable points, while blue points indicate non-scannable points.

learning-based NLOS imaging methods [7–10] enable quick inference, but they require supervision from a large amount of paired data, including well-aligned ground-truth albedos, which are difficult and expensive to acquire. Therefore, exploring NLOS imaging under unsupervised learning is worthwhile to eliminate the heavy reliance on paired data.

In this paper, we propose a novel unsupervised framework capable of learning implicit priors from noisy IUT. This framework comprises two components: a virtual scanning reconstruction network (VSRnet) that learns high-quality measurement-to-albedo mapping beyond the range space induced by the measurement consistency, and a SURE-based denoiser that enhances our method's robustness to measurement noise by incorporating the physics model of the low-photon time-resolved detector. We conduct extensive experiments on simulated data, publicly available real data, and data acquired from our self-built NLOS system. Our method outperforms existing algorithms, particularly in providing robustness for real data and diverse irregular relay surfaces.

Our main contributions are summarized as follows:

• We propose an unsupervised NLOS imaging framework capable of learning implicit priors from noisy IUT, effectively overcoming the dependency on paired data that is difficult to acquire and align.

• We introduce a virtual scanning process that enables the network to learn within both range and null spaces for high-quality reconstruction from IUT, extending NLOS imaging to realistic scenarios with irregular relay surfaces.

• We propose a SURE-based denoiser, an unsupervised physics-guided module that incorporates the low-photon time-resolved detector's physics model, to enhance our method's robustness to noise.

• We evaluate our method on simulated, publicly available, and self-captured real data, demonstrating its superior reconstruction quality and significantly faster inference compared to the SOTA method.

## 2  Related work

### 2.1  Model-based NLOS reconstruction

Model-based NLOS algorithms have achieved significant advances in recent years. Direct reconstruction algorithms [11–15] offer rapid implementations for NLOS imaging, while iterative algorithms [16–18, 5] leverage more accurate physical models for higher reconstruction quality. Recent efforts aim to extend NLOS imaging to challenging real-world scenarios. For instance, Manna et al. [19] and Gu et al. [20] addressed NLOS imaging with dynamic and non-planar relay surfaces, respectively. However, these methods still rely on large relay surfaces and dense measurements. Another set of algorithms [21–25] aims to achieve high spatial resolution reconstruction using sparse sampling measurements to significantly reduce acquisition time. Yet, they are constrained to specific scanning patterns, such as regular or Hadamard patterns. To address NLOS imaging from irregularly undersampled transients, Liu et al. [6] introduced a reconstruction model using Confocal Complemented Signal-Object Collaborative Regularization (CC-SOCR). Despite its effectiveness, CC-SOCR suffers from long inference times due to its iterative nature.

## 2.2 Learning-based NLOS reconstruction

Recently, deep learning has gained attention for NLOS imaging due to its learning capabilities and fast inference. Chopite et al. [7] first employed deep learning for NLOS reconstruction, but their method fell short compared to model-based approaches due to the lack of physical guidance. Physics-guided methods [8, 26, 9, 10] have since emerged to enhance reconstruction quality, particularly for real-world data. Recent works focus on NLOS imaging from regularly undersampled transients [27, 28], but they require large datasets of paired data and fail to reconstruct from IUT. Furthermore, these supervised algorithms still have significant room for improvement in robustness on real data due to the gap between simulated datasets used for training and real-world data.

# 3 Problem formulation and motivation

## 3.1 NLOS Imaging from IUT

Fig. 1 illustrates confocal NLOS imaging with time-resolved systems. By scanning a set of points $P = \{(x_i, y_i, 0) \mid i = 1, 2, \ldots, s, \ x_i, y_i \in \mathbb{R}\}$ on the relay surface, the forward model of NLOS imaging can be modeled as

$$\tau(p, t) = \int_Q \frac{\kappa(q)}{\|p - q\|^4} \cdot \delta(2\|p - q\| - tc)dq \tag{1}$$

where $\tau$ is the spatial-temporal measurement, $p = (x, y, 0)$ denotes the scanning point, $\kappa(q)$ denotes the albedo value at point $q$ in the 3D hidden scene $Q$, $c$ is the speed of light. The distance $\|p - q\|$ is related to the time of flight $t$ through the Dirac delta function $\delta(\cdot)$. For compact presentation and analysis, we employed a discretized version of the above forward model. Let $u \in \mathbb{R}^{st}$ and $\rho \in \mathbb{R}^{l^2 z}$ represent the vectorized measurements and albedos of the hidden object, respectively, where $l$ denotes the size of the vertical and horizontal dimensions, and $z$ denotes the depth dimension. We denote the forward operator, also known as the light transport matrix, by $H \in \mathbb{R}^{st \times l^2 z}$. The forward processing can be described by the following linear model:

$$u = H\rho. \tag{2}$$

Notably, the forward operator $H$ is related not only to the optical-electronic characteristics of the NLOS system but also to the scannable region on the relay surface.

## 3.2 Motivation

Deep learning-based algorithms have demonstrated significant potential to enhance NLOS imaging performance compared to model-based approaches. Most prior work has adopted a supervised paradigm, which requires substantial amounts of high-quality paired data. However, it is prohibitively expensive or even infeasible to acquire ground-truth 3D albedo volumes precisely aligned with the spatio-temporal measurements. This limitation motivates us to develop an unsupervised NLOS reconstruction framework that avoids this dependency and further enhances the generalization of learning-based methods to real-world data.

The standard approach in unsupervised learning is to train the reconstruction mapping $f_\theta$ by minimizing the measurement consistency (MC) loss:

$$\mathbb{E}_u \|H f_\theta(u) - u\|_2^2. \tag{3}$$

Nevertheless, solely enforcing the MC loss without ground truth supervision does not guarantee high-quality reconstruction. This can be analyzed from the perspective of the range-null decomposition [29]. Let $\mathcal{N}_H = \{v \in \mathbb{R}^{l^2 z} \mid Hv = 0\}$ be the null space of the operator $H$. Its complementary space is the range space of $H^\top$, denoted by $\mathcal{R}_H = \{H^\top u, u \in \mathbb{R}^{st}\}$, such that $\mathbb{R}^{l^2 z} = \mathcal{R}_H \oplus \mathcal{N}_H$.

Any albedo volume $\rho$ can be decomposed into a range-space component and a null-space component.

$$\rho = \underbrace{H^\dagger H \rho}_{\text{range-space component}} + \underbrace{(I - H^\dagger H)\rho}_{\text{null-space component}} \tag{4}$$

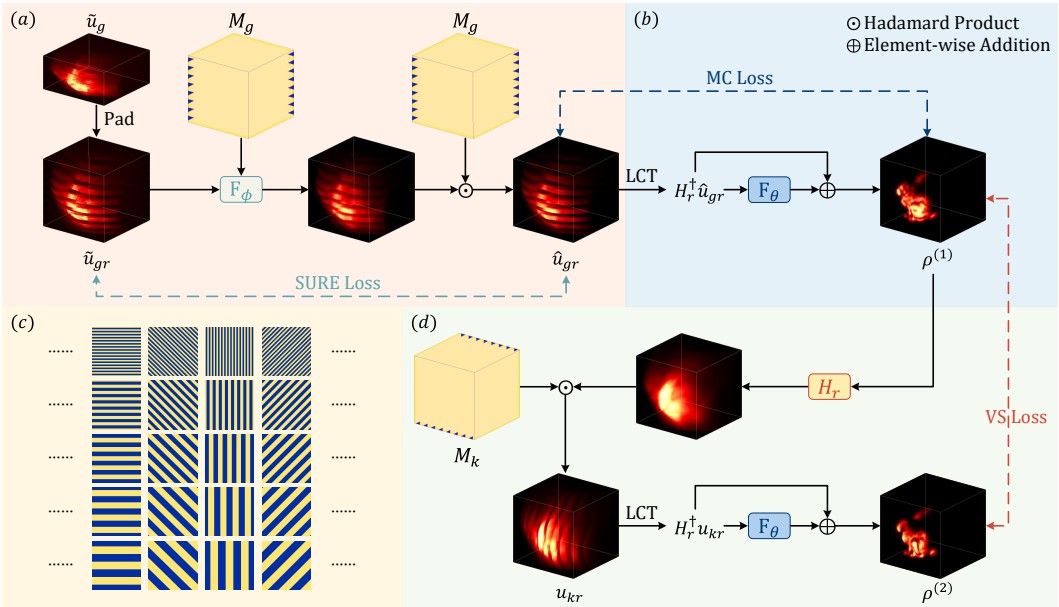

Figure 2: The pipeline of our unsupervised framework: (a) The SURE-based denoiser, which consists of an encoder-decoder network designed for IUT and is trained by minimizing SURE loss in the first stage; (b) The virtual scanning reconstruction network (VSRnet), which consists of a Unet-like network and is trained by MC loss and VS loss in the second stage; (c) Relay surfaces used for training (yellow indicates scanning areas, and their corresponding 3D binary masks are used in implementation); (d) The virtual scanning process, which involves virtually observing $\rho^{(1)}$ with a relay surface $M_k$ that is distinct from $M_g$ and enforcing consistency between $\rho^{(1)}$ and $\rho^{(2)}$.

where $H^\dagger \in \mathbb{R}^{l^2 z \times st}$ is the pseudo-inverse of $H$ satisfying $HH^\dagger H = H$. The operator $H^\dagger H$ projects the sample $\rho$ into the range space $\mathcal{R}_H$: $\mathcal{D}_r(\rho) = H^\dagger H \rho$, whereas its complementary operator $(I - H^\dagger H)$ projects $\rho$ into the null-space $\mathcal{N}_H$: $\mathcal{D}_n(\rho) = (I - H^\dagger H)\rho$. As long as the trained network $f_\theta$ reconstructs from input $u$ as $f_\theta(u) = \mathcal{D}_r(\rho) + v_n$ for $\forall\ v_n\ \in \mathcal{N}_H$, the reconstructed volume $f_\theta(u)$ would fully meet the MC requirement: $H(\mathcal{D}_r(\rho) + v_n) = HH^\dagger H \rho + H v_n = u$ since we have $HH^\dagger H \rho = u$ and $H v_n = 0$. This suggests that the MC constraint only locates the albedo volume in a broad subspace surrounding the range-space projection $\mathcal{D}_r(\rho)$, and the inference of the component $\mathcal{N}_H$ is ad-hoc without guidance.

This necessitates an unsupervised framework capable of learning beyond the range space. We note that model-based algorithms [18, 6, 25, 23] manually design regularization terms to learn beyond range space, but suffer from long inference times. In other computational imaging tasks, supervised methods [30, 31, 29, 32] and unsupervised methods [33–37] have been proposed to recover null-space components of the reconstructions. Along this avenue, we propose an effective unsupervised framework capable of learning in both range-null spaces for NLOS imaging from IUT.

## 4   Method

### 4.1   Unsupervised framework

Fig. 2 shows the proposed unsupervised framework via virtual scanning for NLOS imaging from IUT. The framework consists of two components: 1) a virtual scanning reconstruction network (VSRnet) to recover the 3D albedo volume from both range and null space, and 2) a SURE-based denoiser to enhance the robustness to ubiquitous noise in transients. Given a set of $G$ noisy irregularly undersampled transients $\mathcal{U} = \{\tilde{u}_g \in \mathbb{R}^{st} | g = 1, \ldots, G\}$ and the set of their corresponding forward operators $\mathcal{H} = \{H_g \in \mathbb{R}^{st \times l^2 z} | g = 1, \ldots, G\}$, our goal is to train a deep neural mapping without labeled supervision to reconstruct the 3D albedo volume $\rho$ from the noisy IUT $\tilde{u}$. As discussed in

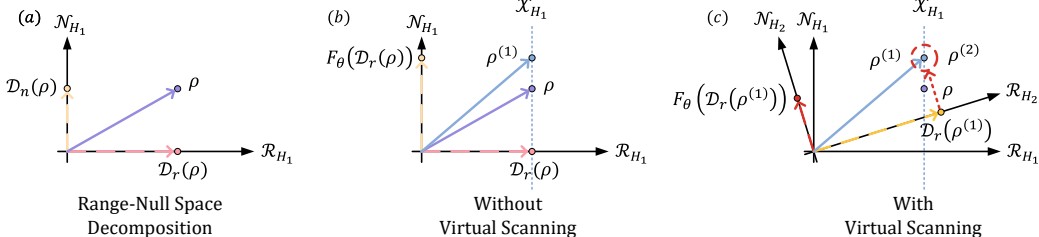

Figure 3: Toy visualization of NLOS reconstruction from the perspective of range-null space decomposition (RNSD). (a) illustrates the RNSD of $\rho$ observed by $H_1$. (b) illustrates that $\rho$ cannot be accurately recovered with only the measurement consistency loss. (c) shows that the proposed virtual scanning promotes the acquisition of null-space components.

Sec. 3.1, due to the one-to-one correspondence between the forward operator and the relay surface, we will loosely use $H_g$ to refer to different relay surfaces without ambiguity in the following sections.

We first briefly sketch the workflow of the inference stage, as shown in Fig. 2(a) and 2(b). The noisy IUT $\tilde{u}_g$, observed through an irregular relay surface $H_g$, is first zero-padded to $\tilde{u}_{gr}$ on a full scanning grid. This is then passed through the SURE-based denoiser $F_\phi$ to remove noise. Subsequently, the albedo volume $\rho$ is reconstructed using VSRnet $f_\theta$, which incorporates the physical prior of NLOS imaging (LCT [11]) and a learned reconstruction mapping $F_\theta$. During the training stage, the denoiser is regularized by the SURE loss, while the reconstruction network is regularized by the MC loss. Additionally, we introduce the virtual scanning process (Fig. 2(d)) to capture the measurement details within the null space of observation operators. In this process, the reconstructed $\rho^{(1)}$ is projected into the measurement space as $u_{kr}$ by virtually scanning with a different relay surface $H_k \in \mathcal{H}$, which is distinct from $H_g$. The virtual scanned measurement $u_{kr}$ is then projected back into the reconstruction space as $\rho^{(2)}$ by the reconstruction network. We impose a virtual scanning (VS) loss between the two reconstructed volumes, $\rho^{(1)}$ and $\rho^{(2)}$, to promote null-space learning (Sec. 4.2). The modules of our framework and the loss functions utilized for training are detailed in the following subsections. The structures of $F_\phi$ and $F_\theta$ are detailed in the Supplementary Material (SM).

## 4.2 Virtual Scanning

**Strategy** The training strategy of VSRnet is depicted in Fig. 2(b) and 2(d). Specifically, at the reconstruction module (Fig. 2(b)), the denoised sample $\hat{u}_{gr}$ is initially transformed to the albedo domain using the inverse operator of LCT $H_r^\dagger$. The resulting $H_r^\dagger \hat{u}_{gr}$ is then mapped to the albedo volume $\rho^{(1)}$ by the reconstruction network $F_\theta$ with a global residual connection. At the virtual scanning module, the reconstructed albedo volume $\rho^{(1)}$ is projected into a virtual undersampled measurement $u_{kr}$ using another forward operator $H_k \in \mathcal{H}$ ($H_k \neq H_g$). To achieve efficient implementation of forward operators, we decouple $H_k$ into $M_k \odot H_r$, which can be efficiently computed using Hadamard product and the fast Fourier transform. In practice, $M_k \in \mathbb{R}^{l \times l \times z}$ is the 3D binary mask associated with the relay surface, constructed by repeating the 2D sampling pattern $t$ times along the time dimension. $H_r$ represents the forward operator of LCT. Following the same reconstruction pipeline, an albedo volume, denoted by $\rho^{(2)}$, is obtained from the virtual measurement $u_{kr}$. The processes of obtaining $\rho^{(1)}$ and $\rho^{(2)}$ can be formalized as:

$$
\begin{aligned}
\rho^{(1)} &= f_\theta(F_\phi(\text{Pad}(\tilde{u}_g), M_g) \odot M_g), \\
\rho^{(2)} &= f_\theta(H_r \rho^{(1)} \odot M_k).
\end{aligned}
\tag{5}
$$

The two reconstructed albedos, $\rho^{(1)}$ and $\rho^{(2)}$, should be identical if the learned mapping $F_\theta$ enables perfect reconstruction. This motivates us to impose a proximity constraint between $\rho^{(1)}$ and $\rho^{(2)}$ in the albedo domain, named the virtual scanning loss. The virtual scanning process in the training pipeline facilitates the learning of the null-space component, thereby providing a promising prior that complements the range space, as analyzed below.

**Analysis** Fig. 3 provides a more intuitive understanding of how the proposed virtual scanning facilitates the network in learning the null-space component. Let $H_1$ and $H_2$ be two observation

operators associated with two different relay surfaces. We observe $\rho$ using the forward operator $H_1$, obtaining the measurement $u_1$. As illustrated in Fig. 3(a), $\rho$ is projected into the range-space $\mathcal{R}_{H_1}$ of $H_1$. When we only impose the MC constraint, the resulting output $\rho^{(1)}$ will belong to the following set $\mathcal{X}_{H_1}$:

$$\mathcal{X}_{H_1} = \{v \mid H_1^\dagger H_1 v = \mathcal{D}_r(\rho), \mathcal{D}_r(\rho) \in \mathcal{R}_{H_1}\}. \tag{6}$$

As depicted by the blue dashed line in Fig. 3(b), there exist multiple outputs that satisfy the measurement consistency. If, for instance, we obtain inaccurate estimated results denoted as $\rho^{(1)}$ through $\rho^{(1)} = \mathcal{D}_r(\rho) + F_\theta(\mathcal{D}_r(\rho))$, we then utilize $H_2$ to virtually scan $\rho^{(1)}$ and project it into the range space $\mathcal{R}_{H_2}$. Subject to the constraint $\rho^{(1)} = \rho^{(2)}$, $F_\theta(\mathcal{D}_r(\rho^{(1)}))$ converges to $\mathcal{D}_n(\rho^{(1)})$ due to the relationship $\rho^{(1)} = \mathcal{D}_r(\rho^{(1)}) + \mathcal{D}_n(\rho^{(1)})$. Following this iteration, the network can learn within $\mathcal{R}_{H_1}$ and $\mathcal{N}_{H_2}$. The entire process is illustrated in Fig. 3(c). Similarly, by altering the order of operators $H_1$ and $H_2$, $F_\theta$ will also observe within $\mathcal{R}_{H_2}$ and $\mathcal{N}_{H_1}$. In practice, a set of operators $\mathcal{H}$ is provided to enhance the network's generalization across various operators with different relay surfaces.

### 4.3 SURE-based denoiser

NLOS imaging inherently operates under photon-limited conditions, and noise in transient measurements can lead to severe background artifacts in the albedo space. This hinders the network's ability to learn implicit priors from noisy IUT, especially in unsupervised learning. Inspired by previous studies [38–42], we introduce a physics-guided unsupervised denoiser to suppress measurement noise. Specifically, we leverage Stein's Unbiased Risk Estimator (SURE) [43] to derive an unsupervised learning loss function that considers the physical model of the time-resolved detector in NLOS imaging systems, which can be modeled as

$$\begin{aligned} \tilde{u} &\sim \text{Poisson}(u + b), \\ u &= H\rho, \end{aligned} \tag{7}$$

where $b$ represents the dark counts of the detector along with the background photons. The parameterized denoiser $F_\phi$ learns to map the noisy measurement $\tilde{u}$ to its clean version $u$ via minimization of the SURE loss function given in Eq. (8).

### 4.4 Loss function

Given a measurement set $\{\tilde{u}_{i,g}, i = 1, 2, \ldots, I, g = 1, 2, \ldots, G\}$ collected by observing $I$ hidden scenes, each with $G$ relay surfaces, the training of our framework involves three loss functions: the SURE loss $\mathcal{L}_{\text{SURE}}$, the MC loss $\mathcal{L}_{\text{MC}}$, and the VS loss $\mathcal{L}_{\text{VS}}$. The SURE loss is an unbiased estimation of the mean squared error (MSE) under the Poisson noise model taking dark count consideration:

$$\begin{aligned} \mathcal{L}_{\text{SURE}} = \mathbb{E}_{\{i,g\}} \Big\{ &\frac{1}{st} \|\tilde{u}_{i,g} - F_\phi(\tilde{u}_{i,g})\|_2^2 - \frac{1}{st}(\mathbf{1} + b)^\top \tilde{u}_{i,g} \\ + &\frac{2}{st} b^\top F_\phi(\tilde{u}_{i,g}) + \frac{2}{st\varepsilon}(e_{i,g} \odot \tilde{u}_{i,g})^\top \big(F_\phi(\tilde{u}_{i,g} + \varepsilon e_{i,g}) - F_\phi(\tilde{u}_{i,g})\big) \Big\} \end{aligned} \tag{8}$$

where $\varepsilon$ is a positive number, $e_{i,g} \in \{-1, 1\}^{st}$ is a binary vector, whose elements follow a Bernoulli distribution with equal probability [40], and $\odot$ denotes element-wise multiplication. Detailed derivation of the SURE loss is given in supp. material.

We adopted the mean squared error for $\mathcal{L}_{\text{MC}}$ and $\mathcal{L}_{\text{VS}}$:

$$\begin{aligned} \mathcal{L}_{\text{MC}} &= \mathbb{E}_{\{i,g\}} \|\hat{u}_{i,g} - H_g(f_\theta(\hat{u}_{i,g}))\|_2^2, \\ \mathcal{L}_{\text{VS}} &= \mathbb{E}_{\{i,g\}} \|f_\theta(\hat{u}_{i,g}) - f_\theta(H_k(f_\theta(\hat{u}_{i,g})))\|_2^2. \end{aligned} \tag{9}$$

We first trained the denoiser $F_\phi$ using the SURE loss. Next, we froze the SURE-based denoiser and trained the network $F_\theta$ with a combined loss function, $\mathcal{L}(\theta) = \mathcal{L}_{MC}(\theta) + \beta \mathcal{L}_{VS}(\theta)$, where $\beta$ is a trade-off parameter (see SM for training details).

## 5 Experiment

### 5.1 Experiment setup

**Dataset** We generated 8,000 transients using the transient rasterizer from [8] with default parameters. The transients have a spatial-temporal resolution of $128 \times 128 \times 512$ with a bin width of 33 ps. The

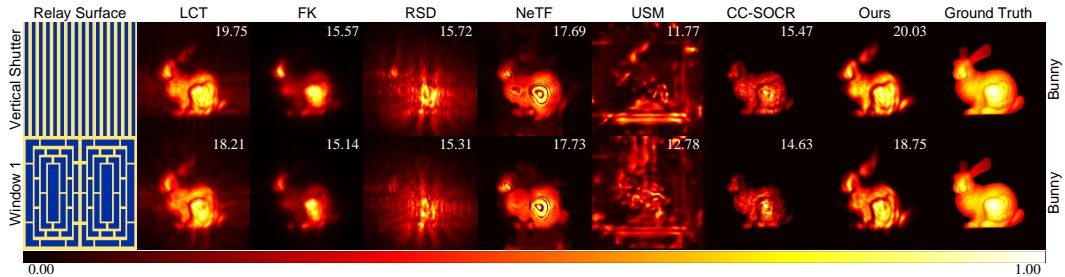

Figure 4: Reconstruction results of the "bunny" with different relay surfaces. The intensity images are normalized to the range of 0 to 1 through maximum value normalization. PSNR (dB) for each intensity image is displayed in the top right corner.

dataset contains 111 objects from the alphanumerics dataset [9], covering lowercase and uppercase letters from the English and Greek alphabets, and numerals from 0 to 9. We also rendered a sample "bunny" for quantitative comparison. To assess our method's generalization, we tested it on real-world data acquired by three different systems [10–12] and a self-built system (see SM for system details). The hidden scenes feature various reflective materials, depth ranges, and geometric shapes, and were acquired under different conditions including scanned areas, spatial resolutions, bin widths, and integration times. For a fair comparison with CC-SOCR [6] within a manageable time frame, we resized the full-sampled transients [10, 12] to $128 \times 128 \times 256$ and the full-sampled transients [11] to $64 \times 64 \times 256$. We used these resized transients for all methods.

**Relay surface**    To simulate the irregularly undersampled process, we extracted signals from the full-sampled transients according to various irregular relay surfaces. For training, we sampled five horizontal shutter patterns with intervals of [4, 8, 12, 16, 20] and 40 uniform rotations from 0 to 180 degrees, resulting in 200 sampling patterns (see Fig. 2(c)). For testing, we included more realistic irregular relay surfaces to evaluate our method's generalization capability in real-world scenarios.

**Compared methods**    We compare our method with three traditional direct reconstruction algorithms (LCT [11], FK [12], RSD [13]), two learning-based algorithms (Unsupervised NeTF [26], Supervised USM [28]), and one iterative algorithm (CC-SOCR [6]). Since LCT, FK, RSD, and USM are designed for raster scanning, we use zero-padded versions of the IUT as input, similar to our method. NeTF and CC-SOCR accept irregularly undersampled transients. For a fair comparison, we also applied our SURE-based denoiser to pre-denoise the transients for all compared methods. However, we did not apply the pre-denoising step for CC-SOCR, as the results reconstructed by CC-SOCR did not show significant improvements. This is because the strong CC-SOCR regularization inherently performs some level of denoising. We compute the peak signal-to-noise ratio (PSNR) to quantitatively evaluate the reconstruction results on the simulated dataset for all methods.

## 5.2   Results

**Simulated data**    Fig. 4 shows comparison results on the "bunny". LCT, FK, NeTF, CC-SOCR, and our method can recover the main body. However, FK and CC-SOCR lose structures around the ear, while LCT and RSD exhibit aliasing artifacts due to irregular undersampling. Among the learning-based methods, NeTF produces a blurry object with diffused artifacts and loss of structures around the ear. USM struggles to adapt to irregularly undersampled transients and fails to reconstruct the hidden object. In contrast, our method successfully recovers most geometric structures of the bunny without aliasing artifacts, achieving the best quantitative results.

**Real-world data**    We first tested our method on publicly available real datasets [10–12] (Fig.5), and then on self-captured real-world data (Fig.6). Without proper regularization, the direct reconstruction methods, *i. e.*, LCT and RSD, exhibit severe aliasing artifacts due to undersampling. FK is depth-sensitive and fails to recover far-end structures in the hidden scene. NeTF tends to produce blurry shapes due to its struggle with utilizing limited information in IUT. USM produces results that are nearly overwhelmed by aliasing artifacts in irregularly undersampled cases, but can achieve acceptable results in regularly undersampled cases (Fig.6). CC-SOCR can recover main objects for

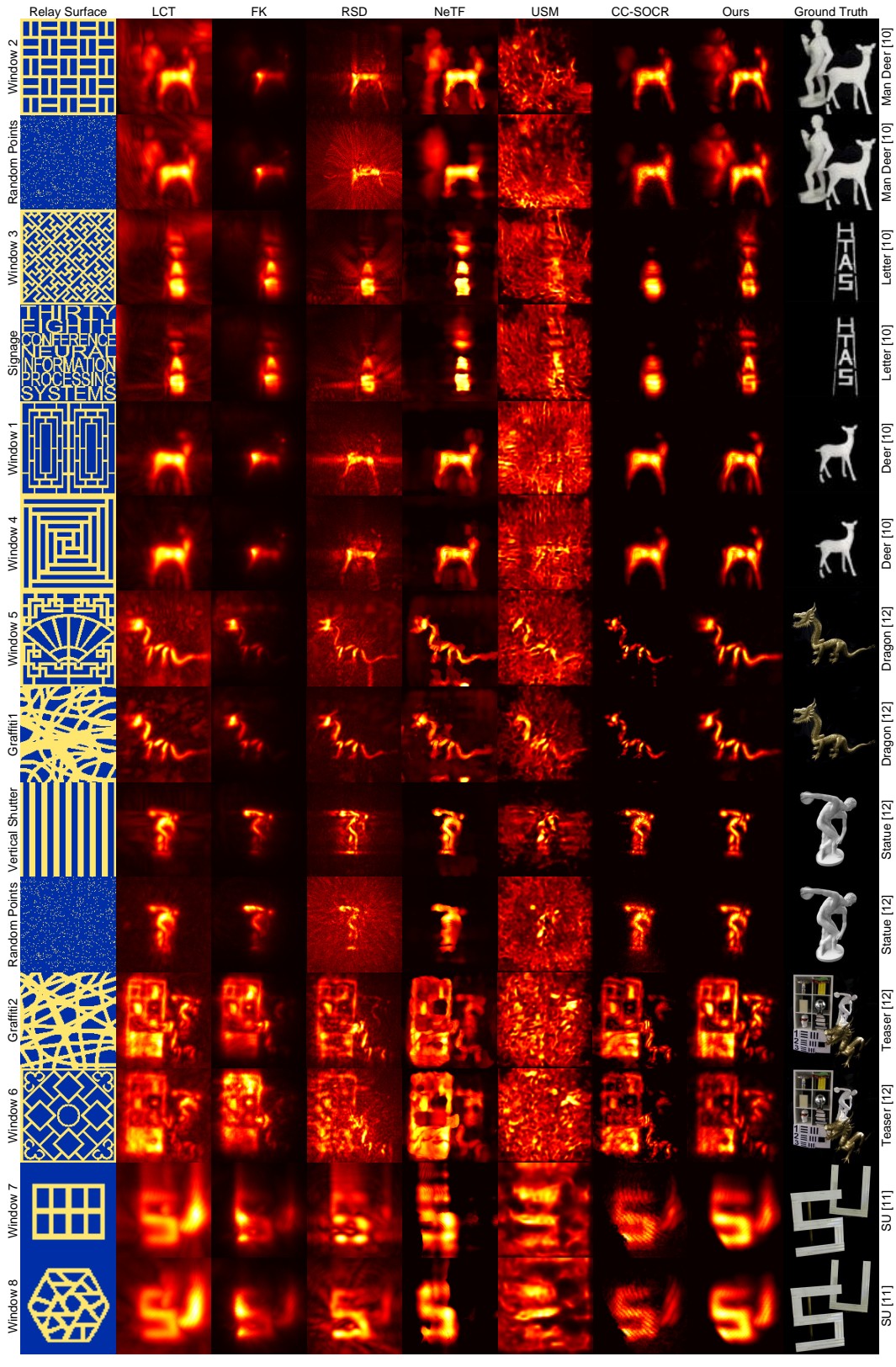

Figure 5: Reconstruction results of publicly available real-world dataset [10–12] with different relay surfaces.

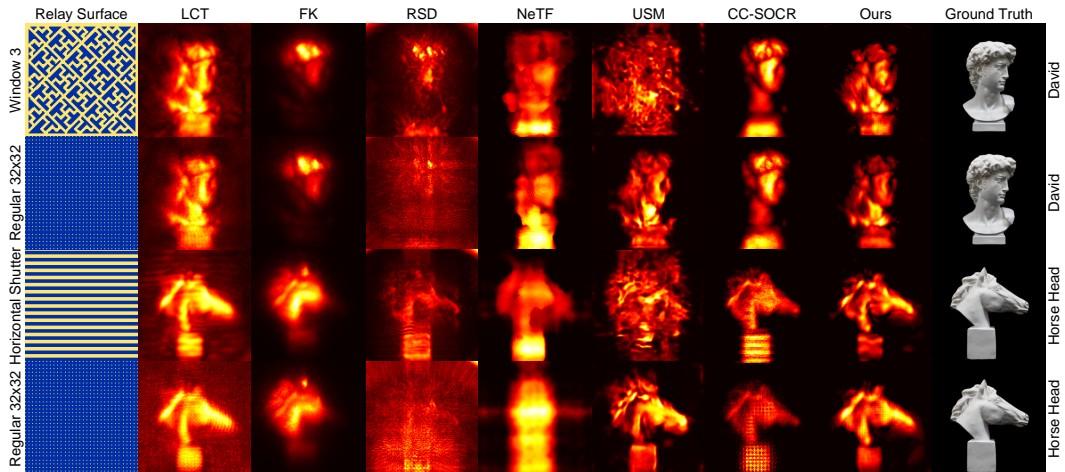

Figure 6: Reconstruction results of self-captured real-world dataset with different relay surfaces.

all the test cases, but loses fine structures at distant depth layers ("Man Deer", "Letter", "Deer" and "Teaser") and underperforms on glossy ("Dragon") or retroreflective ("SU") objects. Our method achieves the best quality for various objects and sampling patterns. Our method generates stable results on real-world data with diverse attributes and relay surfaces despite being trained only on a simple alphanumeric dataset and shutter-like relay surfaces. It notably outperforms the range-space solver LCT, successfully removing aliasing artifacts while preserving structure. The promising generalization highlights the effectiveness of our method in learning beyond the range space.

## 5.3 Inference time

Table 1: Inference time of various methods. The values in the table represent the average inference time across 16 IUT with size of $128 \times 128 \times 256$.

| Method | LCT | FK | RSD | NeTF | USM | CC-SOCR | Ours |
|---|---|---|---|---|---|---|---|
| Runtime (CPU) | 0.81 s | 1.52 s | 0.94 s | N/A | 2.34 s | 7.73 h | 2.24 s |
| Runtime (GPU) | 0.09 s | 0.15 s | 0.12 s | 0.69 h | 0.24 s | N/A | 0.18 s |

We compare the inference time of various methods on an Intel(R) Xeon(R) Platinum 8369B 2.90GHz CPU with 32 cores and an NVIDIA 3090 GPU, respectively. The inference times of NeTF and CC-SOCR vary with different IUT. Therefore, we average the inference times across 16 IUT shown in Fig. 5 and Fig. 6 each with a size of $128 \times 128 \times 256$. As shown in Tab. 1, the direct reconstruction methods, LCT, FK, and RSD, are faster than the other methods. Unlike these one-step approaches, CC-SOCR iteratively solves the functional model by a series of alternative sub-problems, requiring nearly eight hours to reconstruct an albedo. NeTF stands on the per-scene rendering framework and thus requires significantly longer inference times than the other two learning-based methods, USM, and our method. Note that both our method and CC-SOCR achieve more accurate reconstructions than the other methods. However, our method is 12,000× faster than CC-SOCR in CPU mode.

## 5.4 Ablation study

We validate the effectiveness of the two core components of our method: the virtual scanning process (VS) and the SURE-based denoiser. For quantitative evaluation, we simulated a dataset of 1,000 transients by rendering objects with complex geometries, such as chairs, clocks, guitars, sofas, and motorcycles. Our method and its two variants were tested on the simulated transients sampling with 15 different relay surfaces. For qualitative evaluation, we assess the two components using real-world datasets, "Teaser" and "Dragon". "Teaser" features complex structures, which helps evaluate VS's ability to recover details. "Dragon", with its glossy material, exhibits a lower signal-to-noise ratio in its acquired transient, which helps validate the SURE-based denoiser's effectiveness.

Table 2: Ablation study for the virtual scanning process and the SURE-based denoiser.

| SURE-based denoiser | Virtual Scanning Process | PSNR (dB) |
|:---:|:---:|:---:|
| ✗ | ✓ | 18.69 |
| ✓ | ✗ | 19.63 |
| ✓ | ✓ | 20.52 |

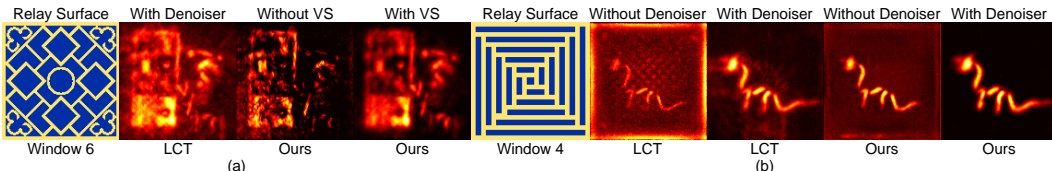

Figure 7: Ablation studies on the virtual scanning process (VS) (a) and the SURE-based denoiser (b).

**Virtual Scanning** As shown in Tab. 2, the virtual scanning process resulted in an average improvement of 0.89 dB. Fig. 7 demonstrates that without the VS component, our method can only recover a limited portion of the scene structure, which is impaired by aliasing artifacts. In contrast, our full model reconstructs more detailed and cleaner structures, confirming the effectiveness of VS component in recovering null-space components.

**SURE-based denoiser** As evident from the quantitative results, the SURE-denoiser achieved an average performance improvement of 1.83 dB, which is more significant than that of VS. This result aligns with expectations because background noise affects the entire reconstruction volume, while aliasing artifacts caused by irregular undersampling disrupt the main structure. For both LCT and our method, as illustrated in Fig. 7(b), the denoiser effectively suppresses noise artifacts. In Sec. 5.2, we apply the denoiser to competing methods to enhance their robustness to noise for a fair comparison. This demonstrates the versatility of the denoiser as a plug-and-play module in other NLOS algorithms.

Additional ablation studies on the hyperparameters within the loss functions and the relay surfaces used for training are provided in the supplementary materials.

## 6 Conclusion and limitations

**Conclusion** In this paper, we propose an unsupervised learning-based framework for NLOS imaging from irregularly undersampled transients (IUT). By introducing a virtual scanning process and a SURE-based denoiser, our framework achieves high-quality and fast NLOS reconstructions from IUT. Furthermore, it can be trained solely from noisy IUT, enabling future work on direct learning from real-world datasets to bridge the gap between simulated and real datasets. Our method outperforms the state-of-the-art method on both simulated and real-world data with various relay surfaces. In future work, we will extend our method to non-confocal imaging systems for more practical applications.

**Limitations** Our method faces two main limitations. Firstly, it is constrained to the confocal system due to the lack of a high-speed, low-memory non-confocal forward operator for deep learning training. However, the framework is theoretically extendable to general forward operators, indicating future research on new forward operators in NLOS imaging may not require special consideration for irregular undersampling. Secondly, while our method is entirely unsupervised, we only present the results of our method which is trained using simulated transients due to the time-consuming nature of collecting real transient datasets. Nonetheless, this approach partially mitigates the data gap introduced by simulated ground truth, and the results demonstrate excellent performance. Transitioning to deep learning that exclusively uses real transients is one of our future goals.

## Acknowledgment

This work is supported by the National Natural Science Foundation of China under Grants 62231018, 62071322, 62072331.

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

# A   Network architecture

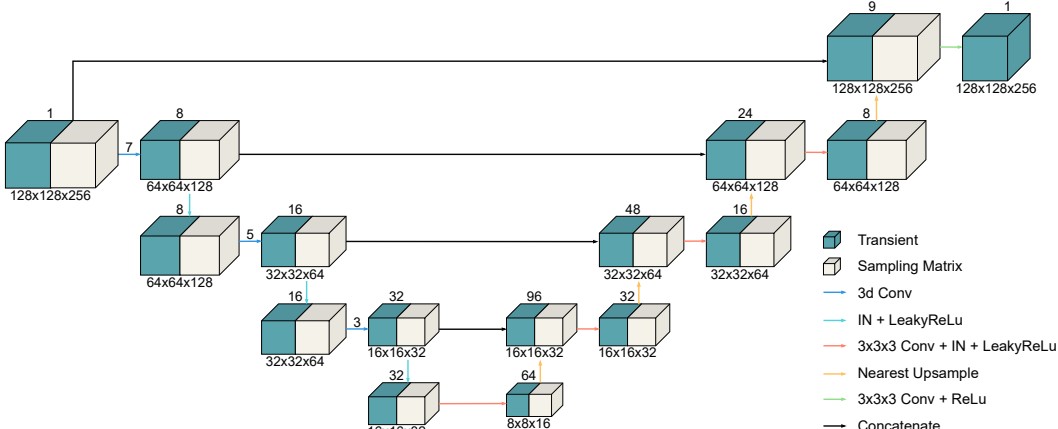

Figure 8: Overview of our P-Unet with 3D partial convolution for denoising irregularly undersampled transients. The spatial resolution of transients features in each layer is depicted below the blocks, and their respective channel counts are presented above the blocks. The resolutions and channels of the sampling matrices are identical to the transients features in the corresponding layer. The numbers on the blue arrows indicate the kernel size of the 3D convolution.

We begin by presenting the architecture of the network $F_\phi$. Since the vanilla convolution treats all voxels of the zero-padded IUT as valid, it leads to the distortion of information in undersampled transients during feature propagation. As illustrated in Fig. 8, we present an encoder-decoder network (P-Unet) based on 3D partial convolution [44]. Partial convolution utilizes a sampling matrix to differentiate between valid and invalid voxels, effectively capturing spatial information from the transients. Additionally, we incorporate the instance normalization (IN) layer [45], which is insensitive to input distribution. This layer enhances the network's generalization capability to diverse zero-padded IUT with varying distributions from different sampling rates.

Given that the performance of $F_\theta$ in VSRnet mainly depends on acquiring null space information from the training forward operators, we adopt a 3D residual network featuring an attention-gated network [46]. While attention mechanisms might not significantly improve reconstruction quality, they help speed up the network's training process.

# B   Training details

Our method is implemented using PyTorch [47], and we employ the Adam optimizer [48] with a weight decay of $10^{-8}$. In the first stage, the SURE-based denoiser model $F_\phi$ is trained with a batch size of 4 for 40 epochs. We set the initial learning rate to $1 \times 10^{-3}$ and reduce it by a factor of 0.1 at epoch 30. Subsequently, in the second stage, the VSRnet model $F_\theta$ is trained with a batch size of 2 for 20 epochs, utilizing an initial learning rate of $5 \times 10^{-4}$ and a reduction by a factor of 0.1 at epoch 10. In each epoch, we randomly select 40 complete simulated transients for each relay surface and extract signals from them to generate irregularly undersampled transients for training. All models were trained on 2 NVIDIA 3090 GPUs, taking nearly 40 hours in total. Regarding the loss function, the hyperparameters $\varepsilon$, $\beta$ and $b$ are set to 0.1, 0.001 and 4, respectively.

# C   Details of proposed SURE-based denoiser

## C.1   Results of compared methods using our denoiser

We applied the proposed SURE-based denoiser to the compared methods and selected relatively better results for comparative experiments. As depicted in Fig. 9, the results of LCT, FK, and RSD exhibit reduced diffuse background noise when the denoiser is applied. For learning-based algorithms, our denoiser effectively removes cluster artifacts caused by measurement noise, thereby enhancing the

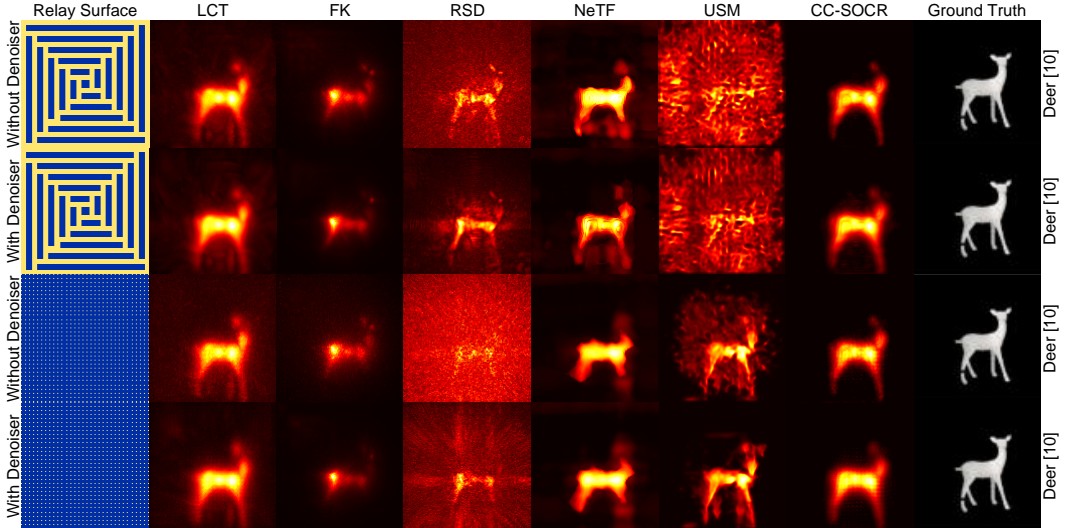

Figure 9: Qualitative results of compared methods with and without the SURE-based denoiser.

clarity of primary structures. However, in the case of CC-SOCR, which already incorporates strong regularization, the improvement of our denoiser is limited. This is likely due to the algorithm's sub-problem already including a denoising effect.

## C.2 Derivation of the SURE loss

Stein's Unbiased Risk Estimation (SURE) framework has been developed and used in prior signal denoising works [38–42, 49]. These works did not consider the dark current of sensor outputs that might be removed in some image signal pipelines. Therefore, we extend the SURE denoising model to NLOS imaging by incorporating more practical noise characteristics of the detector. In this section, we provide the derivation of the SURE loss mentioned in the main text, following the notations in [40, 41].

For compact presentation, we simply denote the training set of transients by $\{\tilde{u}_j \in \mathbb{R}^{st}, j = 1, 2, \ldots, I \times G\}$ instead of the two-subscript version $\{\tilde{u}_{i,g}, i = 1, 2, \ldots, I, g = 1, 2, \ldots, G\}$. $I$ is the number of observed hidden scenes, and $G$ is the number of forward operators associated with relay surfaces. $s$ and $t$ are the number of scanning points and the number of time bins in each histogram of the transient, respectively.

In the supervised learning, given a paired dataset of noisy transients $\{\tilde{u}_j\}$ and its clean version $\{u_j\}$, the model $F_\phi$ can be trained by minimizing the mean squared error (MSE) loss function:

$$\mathbb{E}_{\{\tilde{u},u\}}\left\{ \sum_{j=1}^J \frac{1}{st}\|u_j - F_\phi(\tilde{u}_j)\|^2 \right\} = \mathbb{E}_u\left\{ \sum_{j=1}^J \frac{1}{st}\mathbb{E}_{\tilde{u}|u}\|u_j - F_\phi(\tilde{u}_j)\|^2 \right\}. \tag{10}$$

However, the clean transient set $\{u_j\}$ is not available. To achieve unsupervised learning only with noisy transients, our goal is to obtain an unbiased estimator of the mean squared error (MSE) standing on the SURE framework. To this end, we can further decompose the inner expectation in Eq. 10 as

$$\mathbb{E}_{\tilde{u}|u}\|u_j - F_\phi(\tilde{u}_j)\|^2 = \mathbb{E}_{\tilde{u}|u}\big\{\|u_j\|^2\big\} + \mathbb{E}_{\tilde{u}|u}\big\{\|F_\phi(\tilde{u}_j)\|^2\big\} - 2\mathbb{E}_{\tilde{u}|u}\big\{u_j^\top F_\phi(\tilde{u}_j)\big\}. \tag{11}$$

The unbiased estimator of the second term in Eq. 11 is $\|F_\phi(\tilde{u}_j)\|^2$, which can be obtained without the clean transient $u_j$. The first term in Eq. 11 can be rewritten as $\mathbb{E}_{\tilde{u}|u}\big\{u_j^\top \tilde{u}_j\big\}$ because $\mathbb{E}_{\tilde{u}|u}\big\{\tilde{u}_j\big\} = u_j$. The first and third terms depend on $u$. Thus, we need to obtain unbiased estimators for them by considering the characteristics of the noise model .

In the case of NLOS imaging, the noisy transients can be modeled as $\tilde{u} \sim \text{Poisson}(u + b)$, where $b$ denotes the dark counts. The unsupervised expressions of $\mathbb{E}_{\tilde{u}|u}\big\{u_j^\top \tilde{u}_j\big\}$ and $\mathbb{E}_{\tilde{u}|u}\big\{u_j^\top F_\phi(\tilde{u}_j)\big\}$ can be obtained using the following lemma.

***Lemma* 1 (Lemma 1.2 in [40])**  *Let $v \in \mathbb{R}^{st}$ such that $v \sim Poisson(u)$ be an independent random variables and let $\Phi : \mathbb{R}^{st} \to \mathbb{R}^{st}$ be a function such that $\mathbb{E}_{v|u}\{|\Phi_m(v)|\} < +\infty$ for all $m$.*

$$\mathbb{E}_{v|u}\{u^\top \Phi(v)\} = \mathbb{E}_{v|u}\{v^\top \Phi^{[-1]}(v)\}. \tag{12}$$

Let $\Phi(v) = F_\phi(\tilde{u})$, then $\mathbb{E}_{\tilde{u}|u}\{u_j^\top F_\phi(\tilde{u}_j)\}$ can be expressed as:

$$\begin{aligned}
&\mathbb{E}_{\tilde{u}|u}\{u_j^\top F_\phi(\tilde{u}_j)\} \\
&= \mathbb{E}_{v|u}\{u_j^\top \Phi(v_j)\} \\
&= \mathbb{E}_{v|u}\{(u_j + b)^\top \Phi(v_j)\} - \mathbb{E}_{v|u}\{b^\top \Phi(v_j)\} \\
&= \mathbb{E}_{v|u}\{v_j^\top \Phi^{[-1]}(v_j)\} - \mathbb{E}_{v|u}\{b^\top \Phi(v_j)\} \\
&= \mathbb{E}_{\tilde{u}|u}\{\tilde{u}_j^\top F_\phi^{[-1]}(\tilde{u}_j)\} - \mathbb{E}_{\tilde{u}|u}\{b^\top F_\phi(\tilde{u}_j)\}.
\end{aligned} \tag{13}$$

Here, we use the first-order Taylor approximation $F_\phi^{[-1]}(\tilde{u}_j) \approx F_\phi(\tilde{u}_j) - \partial F_\phi(\tilde{u}_j)$ to simplify Eq. 13 into:

$$\mathbb{E}_{\tilde{u}|u}\{\tilde{u}_j^\top F_\phi(\tilde{u}_j)\} - \mathbb{E}_{\tilde{u}|u}\{\tilde{u}_j^\top \partial F_\phi(\tilde{u}_j)\} - \mathbb{E}_{\tilde{u}|u}\{b^\top F_\phi(\tilde{u}_j)\}. \tag{14}$$

Then, the unbiased estimator of Eq. 14 is

$$\tilde{u}_j^\top F_\phi(\tilde{u}_j) - \tilde{u}_j^\top \partial F_\phi(\tilde{u}_j) - b^\top F_\phi(\tilde{u}_j). \tag{15}$$

Note that it is difficult to obtain an analytic form of $\partial F_\phi(\tilde{u}_j)$ when $F_\phi(\cdot)$ is a neural network. We adopt the Monte-Carlo approach [49] to obtain an estimate of the divergence of $F_\phi(\tilde{u})$ with the following approximation:

$$div_{\tilde{u}}\{F_\phi(\tilde{u})\} \approx \frac{e^\top}{\varepsilon}(F_\phi(\tilde{u} + \varepsilon e) - F_\phi(\tilde{u})), \tag{16}$$

where $\varepsilon$ is a small positive number, and $e \in \{-1, 1\}^{st}$ is a binary vector whose entities follow a Bernoulli distribution with equal probability [40].

Similarly, let $\Phi(v)$ be an identity function such that $\Phi(v) = v$, and $v = \tilde{u}$. Using the first-order Taylor approximation, $\mathbb{E}_{\tilde{u}|u}\{u_j^\top \tilde{u}_j\}$ can be expressed as

$$\begin{aligned}
&\mathbb{E}_{\tilde{u}|u}\{u_j^\top \tilde{u}_j\} \\
&= \mathbb{E}_{v|u}\{v_j^\top \Phi^{[-1]}(v_j)\} - \mathbb{E}_{v|u}\{b^\top \Phi(v_j)\} \\
&\approx \mathbb{E}_{v|u}\{v_j^\top (v_j - \partial v_j)\} - \mathbb{E}_{v|u}\{b^\top \Phi(v_j)\} \\
&= \mathbb{E}_{\tilde{u}|u}\{\tilde{u}_j^\top (\tilde{u}_j - \partial \tilde{u}_j)\} - \mathbb{E}_{\tilde{u}|u}\{b^\top \tilde{u}_j\}.
\end{aligned} \tag{17}$$

We obtain the unbiased estimator of Eq. 17:

$$\tilde{u}_j^\top (\tilde{u}_j - \mathbf{1}) - b^\top \tilde{u}_j, \tag{18}$$

where $\mathbf{1}$ is a vector consisting of $st$ ones. With the above derivation, the total unbiased estimator of the MSE is given by:

$$\begin{aligned}
\sum_{j=1}^{J} \frac{1}{st}\Big\{ &\|\tilde{u}_j\|^2 - 2\tilde{u}_j^\top F_\phi(\tilde{u}_j) + \|F_\phi(\tilde{u}_j)\|^2 - \mathbf{1}^\top \tilde{u}_j - b^\top \tilde{u}_j \\
&+ 2b^\top F_\phi(\tilde{u}_j) + \frac{2}{\varepsilon}(e_j \odot \tilde{u}_j)^\top (F_\phi(\tilde{u}_j + \varepsilon e_j) - F_\phi(\tilde{u}_j)) \Big\}.
\end{aligned} \tag{19}$$

After rearrangement, Eq. 19 can be expressed as:

$$\begin{aligned}
\sum_{j=1}^{J} \frac{1}{st}\Big\{ &\|\tilde{u}_j - F_\phi(\tilde{u}_j)\|^2 - (\mathbf{1} + b)^\top \tilde{u}_j \\
&+ 2b^\top F_\phi(\tilde{u}_j) + \frac{2}{\varepsilon}(e_j \odot \tilde{u}_j)^\top (F_\phi(\tilde{u}_j + \varepsilon e_j) - F_\phi(\tilde{u}_j)) \Big\}.
\end{aligned} \tag{20}$$

Thus, we get the SURE loss function as described in the main text:

$$\begin{aligned}
\mathcal{L}_{\text{SURE}} = \mathbb{E}_{\{i,g\}}\Big\{ &\frac{1}{st}\|\tilde{u}_{i,g} - F_\phi(\tilde{u}_{i,g})\|_2^2 - \frac{1}{st}(\mathbf{1} + b)^\top \tilde{u}_{i,g} \\
&+ \frac{2}{st}b^\top F_\phi(\tilde{u}_{i,g}) + \frac{2}{st\varepsilon}(e_{i,g} \odot \tilde{u}_{i,g})^\top (F_\phi(\tilde{u}_{i,g} + \varepsilon e_{i,g}) - F_\phi(\tilde{u}_{i,g})) \Big\}.
\end{aligned} \tag{21}$$

During the training process, we use zero-padded irregularly undersampled transients as input, but only consider valid values in the calculation of the SURE loss.

# D  Details of our NLOS system

We used a femtosecond fiber laser as a light source with a central wavelength of 1560 nm and a pulse repetition rate of 82 MHz. The light was split by a $99 : 1$ fiber coupler. The channel with 1% of the light was attenuated and sent into a fast photodetector, functioning as the start signal for the time-to-amplitude convertor (TAC). The other channel, with 99% of the light, was connected to a collimator by a piece of single mode fiber (SMF) and then travelled through a hole in the mirror. The beam was diffusively reflected by the relay surface, and the raster scanning was controlled by the beam-steering mirror. After diffusive reflections by the object and again by the relay surface, the echo light travelled back along the same optical path and was reflected to a second collimator. Finally, the echo light was detected by a fractal superconducting nanowire single-photon detector (SNSPD) coupled with SMF. The output voltage pulses from the fractal SNSPD were amplified by the RF amplifiers and input to the TAC as a stop signal for the coincidence counting. The transients were captured over a $0.8 \times 0.8 \ m^2$ scanning area, with a size of $128 \times 128 \times 512$ and a bin width of 8 ps.

# E  Discussion

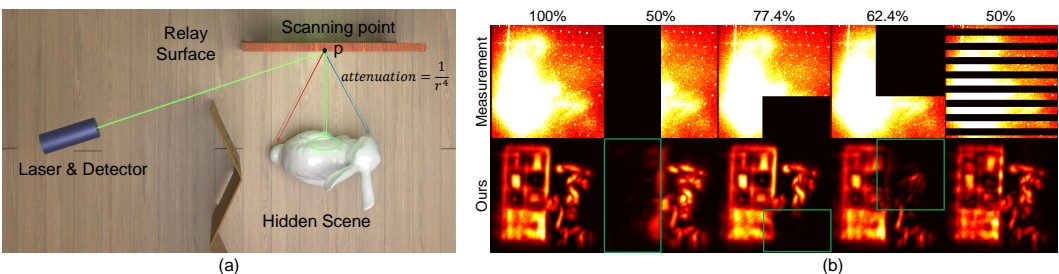

(a)  (b)

Figure 10: (a) The top view of the NLOS imaging system. The green oval region on the object indicates the area from which the scanning point can capture information. Light with different optical distances is represented by three colors. (b) The first row shows the maximum intensity projections of the transient along the time dimension, with the corresponding sampling rates indicated at the top.

In this section, we discuss how different relay surfaces affect reconstruction quality. Generally, a relay surface with a large missing area leads to result in poor reconstruction quality due to information loss. To verify this, we removed signals from the fully-sampled transient according to the relay surfaces and reconstructed them using our method. It is clear that parts of the hidden scene facing large missing areas are nearly impossible to reconstruct, as illustrated in the second, third, and fourth columns of Fig 10 (b). However, even with the same sampling rate, a more uniform distribution of scanning points produces better results (e.g., comparing the fifth column to the second).

Therefore, in the challenging task of NLOS imaging from IUT, the sampling rate is not the sole determinant of reconstruction quality. When the detector focuses a point on the relay surface, only the parts of the hidden scene that are close to that point can be observed. As shown in Fig 10 (a), the light attenuation term $1/r^4$ causes that a histogram ($1 \times 1 \times T$) captured from scanning point $p$ primarily contains information from hidden areas close to that point, as indicated by the green oval area. This results in poor reconstruction when large areas of the relay surface are invalid for scanning. However, determining the acceptable proportion of the invalid area relative to the entire surface for successful reconstruction is theoretically challenging, as it depends on various factors, including the relative depth between the hidden scene and the relay surface, the scene's reflectivity and normals, detector efficiency, and laser power. We consider this an important area for future work.

# F  Additional ablation study

In this section, we extend our ablation study further to explore the optimal values and sensitivity of the hyperparameters within the loss functions. Additionally, we examine the effect of relay surfaces used for training. To accommodate variations in testing data, we evaluated our method on different IUT of the "bunny", sampled according to 15 distinct relay surfaces mentioned in the main text.

### F.1 Ablation study on hyperparameters

Table 3: Effect of the hyperparameters $\varepsilon$ on the denoising performance in terms of PSNR (mean value and variance).

| $\varepsilon$ | 0.0001 | 0.001 | 0.01 |
|---|---|---|---|
| PSNR | 39.11±1.99 | 40.19±2.91 | 41.16±2.61 |
| $\varepsilon$ | **0.1** | 1 | 10 |
| PSNR | **41.53 ± 1.25** | 36.97±7.35 | 34.92±0.17 |

**Effect of the positive number $\varepsilon$**   In our experiments, the transient is normalized to the range $[0, 100]$. Following the recommendation in [50], where the value of $\varepsilon$ is suggested to be set around $\max(u)/1000$, we set $\varepsilon$ to 0.1. As shown in Tab. 3, our SURE-based denoiser exhibits optimal performance when $\varepsilon = 0.1$.

Table 4: Effect of the hyperparameters $\beta$ on the reconstruction performance in terms of PSNR (mean value and variance).

| $\beta$ | 0 | 0.0001 | 0.0005 |
|---|---|---|---|
| PSNR | 16.19±0.21 | 17.36±0.45 | 18.25±0.77 |
| $\beta$ | **0.001** | 0.005 | 0.01 |
| PSNR | **19.03 ± 0.64** | 18.39±0.81 | 18.03±0.86 |

**Effect of the hyperparameter $\beta$**   The hyperparameter $\beta$ acts as a weight to balance the MC loss and the VS loss. As shown in Tab. 4, our method achieves optimal performance when $\beta = 0.001$. The effectiveness of our method decreases notably as $\beta$ decreases. When the VS loss is completely disabled ($\beta = 0$), the model fails to learn beyond the range space, highlighting the significance of virtual scanning in our approach.

### F.2 Ablation study on relay surfaces for training

It's widely recognized that the robustness of neural networks depends heavily on the diversity of the training dataset. As discussed in [51], some neural networks struggle to adapt to changes in the sampling rate. They perform best when the sampling rate of input matches that of training dataset. Any deviation, whether a decrease or increase in the testing data's sampling rate, can lead to a decline in performance. To address this challenge, a common strategy is to augment the diversity of the training data's sampling rate.

Table 5: Effect of the intervals of relay surfaces for training on the reconstruction performance in terms of PSNR (mean value and variance).

| Interval | [4] | [12] | [20] |
|---|---|---|---|
| PSNR | 17.63±0.64 | 17.77±0.95 | 18.16±0.76 |
| Interval | [8,16] | [4,12,20] | **[4,8,12,16,20]** |
| PSNR | 18.36±1.05 | 18.85±0.88 | **19.03 ± 0.64** |

**Interval**   In the context of NLOS imaging from irregularly undersampled transients, we can modify the shape of the relay surface to adjust the sampling rate of transients. In theory, light reflecting from the hidden scene spreads across the entire relay surface. However, due to the radiometric fall-off $\|p - q\|^4$, only a limited portion of the reflected light, concentrated in a small area of the relay surface, can be effectively captured by the time-resolved detector. As a result, we manipulate the interval of the shutter-like surfaces to modify the local sampling rate of undersampled transients.

As shown in Tab. 5, the optimal combination of interval values for relay surfaces is $[4, 8, 12, 16, 20]$. Our method's performance shows a decline with a reduction in the variety of interval types. This suggests that using undersampled transients with a more diverse range of sampling rates for training can enhance the generalization capability of our method.

Table 6: Effect of the rotations of relay surfaces for training on the reconstruction performance in terms of PSNR (mean value and variance).

| Rotation | 10 | 20 | 30 |
|---|---|---|---|
| PSNR | 18.08±0.65 | 18.33±1.05 | 18.74±1.32 |
| Rotation | **40** | 50 | 60 |
| PSNR | **19.03 ± 0.64** | 18.85±0.94 | 18.76±0.75 |

**Rotation** Similarly, we can modulate the diversity of relay surfaces by varying the number of rotations. As shown in Tab. 6, the optimal number of rotations is 40 when the count of transients using for training is 8,000 in total. When the number of rotations falls below 40, the diversity of relay surfaces becomes inadequate. Conversely, when it exceeds 40, the allocated transients for each relay surfaces becomes too limited.

# G   Additional results

We present additional results on publicly available real data [10, 12] with more irregular relay surfaces, as shown in Fig 11 and Fig 12. The experiment setup for data processing and the compared methods remains consistent with the description in the main text.

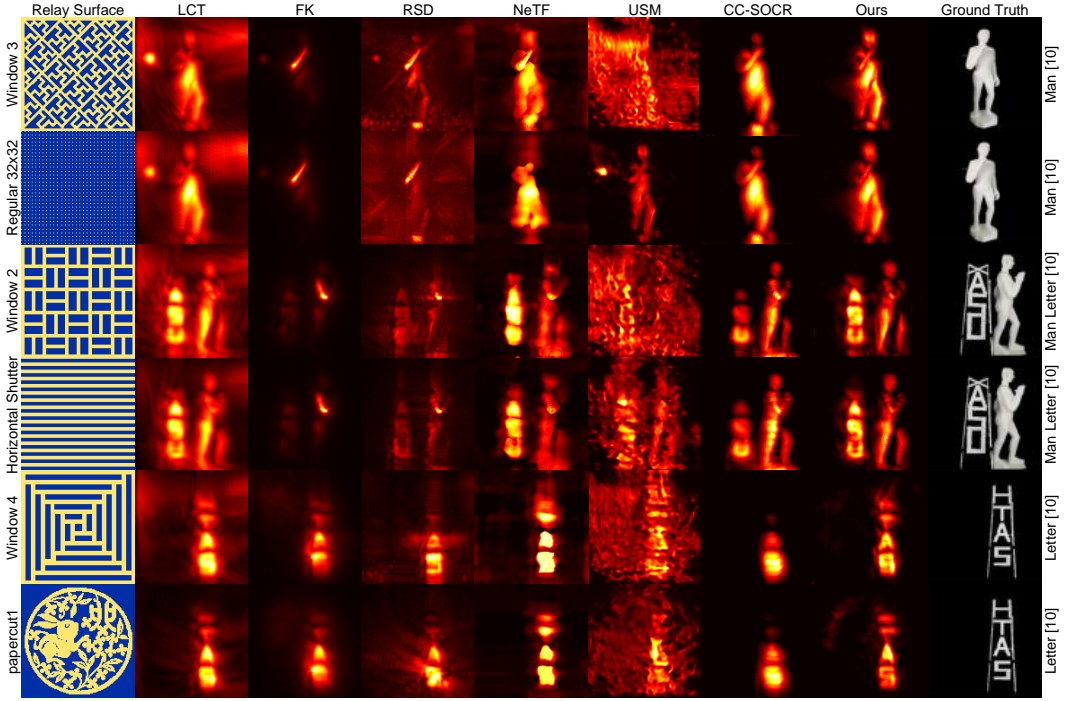

Figure 11: Reconstruction results of publicly available real-world dataset [10] with different relay surfaces.

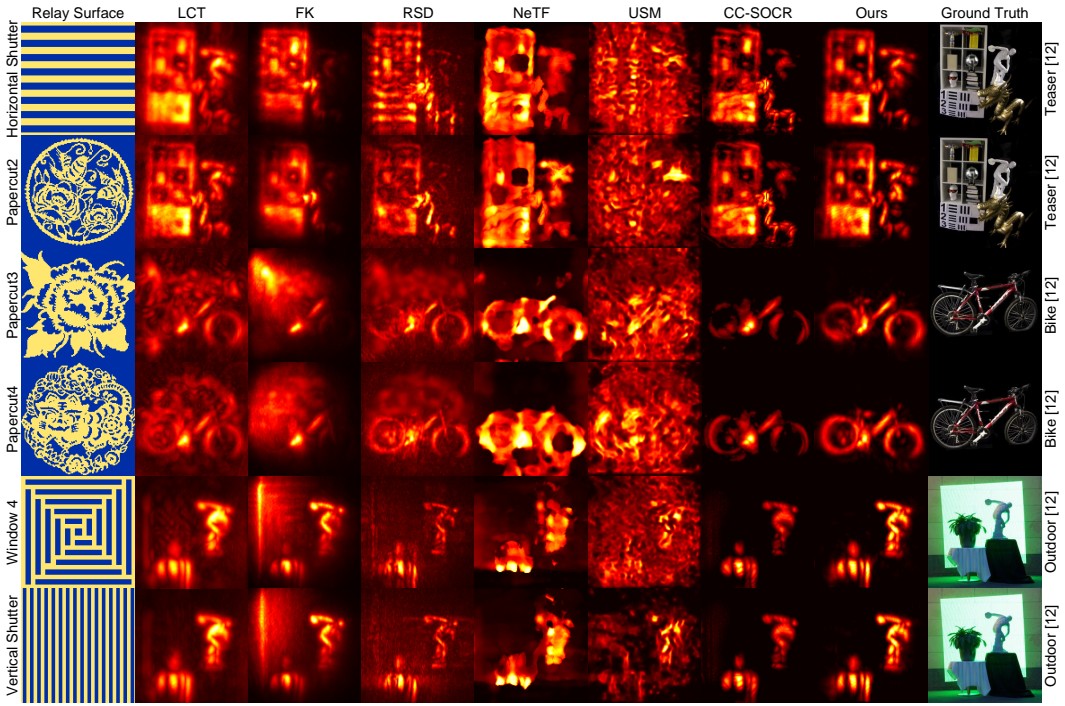

Figure 12: Reconstruction results of publicly available real-world dataset [12] with different relay surfaces.

