# OpenReview forum: "Virtual Scanning: Unsupervised Non-line-of-sight Imaging from Irregularly Undersampled Transients"
_NeurIPS.cc/2024/Conference — NeurIPS 2024 poster_

### Official Review · Reviewer_wGeV · 2024-07-10

**Soundness:** 3
**Presentation:** 3
**Contribution:** 3
**Rating:** 6
**Confidence:** 3

**Summary:**

The paper introduces a unique unsupervised learning framework for non-line-of-sight imaging from irregularly undersampled transients. The proposed method overcomes the dependency on paired data and achieves higher fidelity, greater robustness, and remarkably faster inference times.

**Strengths:**

1. The paper proposes an unsupervised learning framework that overcomes the dependence on dense scanning and paired data required by existing methods.
2. Extensive experiments, including both simulated and real-world data, validate the effectiveness and generalization ability of the method.

**Weaknesses:**

1. The 'irregular windows' in the test scenarios composed of regular, repetitive, or symmetrical simple geometric shapes, which are more close to the relay surfaces simulated during training. Whether this method can be extended to more complex real-world scenarios remains a concern.

**Questions:**

1. Have you considered other irregular relay surfaces that is not composed of simple repetitive or symmetrical shapes?
2. What is the motivation of using different shutter pattens as the simulated relay surfaces? Have you tried other patterns?

I will reconsider my rating if your answers address my confusion.

**Limitations:**

The authors presented the limitations and proposed some future direction to addressed them.

---

> ### Author Rebuttal · Authors · 2024-08-07
>
> We are highly encouraged by the positive recommendation and comments from the reviewer on our experiment, method and presentation. Furthermore, the comments and suggestions are inspiring, helpful, and valuable. We address the main issues as follows.
>
> **Q1: Irregularity of scanning patterns.**
>
> **Reply:**
> We tested our method on publicly available real datasets with more irregular relay surface. These irregular relay surfaces do not contain simple repetitive or symmetrical shapes, such as the signage of NIPS 2024, window papercut with various shapes and even random graffiti. As shown in Figure 3, our method still achieves the best quality for various sampling patterns. In addition, as shown in Fig. 5 and 6 in the manuscript, the relay surface ''random points'' and ''regular 32$\times$32'' are more different from our training patterns, but our method can also handle them well. This is because our method's generalization is guaranteed by the theoretically grounded framework in learning beyond the range space.
>
> **Q2: The motivation of using different shutter patterns for training.**
>
> **Reply:** In the early development of our pipeline, we have tried to use random irregular relay surfaces for training, hoping that it would enhance our algorithm's robustness. However, this setting resulted in dramatic loss fluctuations and unstable training.
> We found that shutter-like patterns yield stable training. By adjusting the rotation angles and intervals of these patterns, the network generalizes well to a wide range of irregular sampling patterns (see Ablation studies in Sec. C.2).

---

> > ### Comment · Reviewer_wGeV · 2024-08-08
> > **My concern is resolved**
> >
> > Thanks for your reply. I would like to raise my rating.

---

> > > ### Author Response · Authors · 2024-08-08
> > > **Response Acknowledgement and Appreciation**
> > >
> > > Thank you for your positive feedback and for raising your rating. We appreciate your thoughtful review and are glad that our response addressed your concern.

---

### Official Review · Reviewer_8wFw · 2024-07-11

**Soundness:** 4
**Presentation:** 3
**Contribution:** 4
**Rating:** 8
**Confidence:** 4

**Summary:**

To address the challenges of slow inference speed and poor generalization to irregularly relay surfaces
in non-line-of-sight (NLOS) imaging, this paper proposes a learnable unsupervised training framework
with the excellent novelty, as well as a Virtual Scanning Reconstruction Network (VSRnet).
Furthermore, to mitigate the noise present in transient data, the authors introduce a denoiser based on
Stein's Unbiased Risk Estimate (SURE) loss. Extensive comparative experiments and ablation studies
demonstrate that the proposed method significantly enhances reconstruction quality compared to
existing approaches.

**Strengths:**

1. The authors provide a novel analysis of the challenges faced by unsupervised methods in NLOS
imaging applications. They specifically incorporate different types of relay surfaces as prior information
during the training phase of the model, which effectively enhances the network's generalization ability to
transient data collected from irregular relay surfaces.
2. Based on relevant mathematical theories, the authors provide detailed derivation steps for the
proposed SURE Loss in the supplementary material. They offer a computable form of the unbiased
estimation term using techniques such as Monte Carlo approximation. Although this loss function is
primarily designed based on existing theoretical work, its application in NLOS imaging remains a
valuable contribution.
3. The authors provide extensive experimental results on real-world data, demonstrating that their
proposed method can deliver high-quality reconstruction results

**Weaknesses:**

1. The lack of explanations for some key symbols (e.g., the explanation for M_g in Figure 2(a)
and the construction method of M_k in Fig. 2(d)) and schematics (e.g., Fig. 3(c) is difficult for
readers to understand from the brief descriptions in the paper) might hinder readers from
reproducing the proposed algorithm. Providing further explanations could significantly enhance
the readability of the paper.
2. There is a lack of quantitative evaluation results for VSRnet and SSIM-denoiser. Section 5.4 of
the paper only provides qualitative comparisons of the two modules, neglecting quantitative
experimental assessments. For readers, quantitative results are more effective than qualitative
results in intuitively understanding the model's effectiveness.
3. The title of the paper includes the keywords "unsupervised" and "irregularly undersampled transients," which seem to represent two independent issues. Specifically, is the choice of an unsupervised framework a necessary outcome for addressing the reconstruction problem of irregularly undersampled transients? If the authors could clarify this point, I believe it would constructively enhance the paper.

**Questions:**

1. I am confused about the content in M_k and M_g. Could you please provide further definitions or
explanations for M_k and M_g?
2. The masks used during training exhibit regular patterns (e.g., evenly spaced stripes). Would
introducing more irregularly varied masks further enhance the robustness of the network?
3. It is unclear how the authors constructed different forward propagation operators H based on the
varying irregularly relay surfaces?

**Limitations:**

Discussed.

---

> ### Author Rebuttal · Authors · 2024-08-07
>
> We are highly encouraged by the positive recommendation and comments from the reviewer. We address the main questions as follows.
>
> **Q1: The lack of explanations for some key symbols and schematics.**
>
> **Reply:**
> Explanations: As stated in line 90-92, the forward operator $H$ is highly related to the scannable region on the relay surface.
> For the relay surface numbered $g$, $H_g$ is the forward operator using this relay surface, $M_g$ is the 3D version mask of the sampling pattern, constructed by repeating the 2D sampling pattern $t$ times along the time-dimension.
> Similarly, $M_k$ is another 3D version mask different from $M_g$.
>
> Schematics:  Figure 3(c) (main text) builds on Figure 3(b) (main text), which shows the training case with only the measurement consistency (MC) loss, corresponding to Figure 2(b) (main text).
> If only the MC constraint is imposed, the network cannot uniquely recover $\rho$ because multiple outputs can satisfy the MC constraint, as depicted by the blue dashed line in Figure 3(b) (main text).
> In Figure 3(c) (main text), when using the virtual scanning, $\rho^{(1)}$ is projected onto the $H_2$'s range-space $R_{H_2}$, and we can recover the range-space component $D_r(\rho^{(1)})$ of $\rho^{(1)}$ using the pseduo-inverse operator (LCT).
> Given $\rho^{(2)}=D_r(\rho^{(1)})+F_\theta(D_r(\rho^{(1)}))$ and $\rho^{(1)}=D_r(\rho^{(1)})+D_n(\rho^{(1)})$, the proposed VS loss $\rho^{(1)}=\rho^{(2)}$ is equivalent to $F_\theta(D_r(\rho^{(1)}))=D_n(\rho^{(1)})$.
> Therefore, virtual scanning ensures that the network $F_\theta$ learns a mapping from $D_r(\rho)$ to $D_n(\rho)$, ensuring the ability of our algorithm to achieve high-quality reconstruction.
>
> **Q2: The quantitative evaluation results for VSRnet and SURE-denoiser.**
>
> **Reply:**
> We simulated a dataset of 1,000 transients by rendering objects (chairs, clocks, guitars, sofas, motorcycles) with random scaling and positioning.
> We tested our method and its variants on the simualted dataset sampling with 16 various relay patterns.
> The quantitative results for VSRnet and SURE-based denoiser are as follows:
> | SURE-denoiser | Virtual Scanning | PSNR (dB) |
> |----------|----------|----------|
> | &times; | &check; | 18.69 |
> | &check; | &times; | 19.63 |
> | &check; | &check; | 20.52 |
>
> As evident from the results, the SURE-denoiser provided an average performance improvement of 1.83 dB and the virtual scanning provided an average improvement of 0.89 dB.
> The SURE-denoiser's quantitative performance improvement is more significant because background noise affects the entire reconstruction volume, while aliasing artifacts caused by irregular undersampling mainly disrupt the primary structure.
>
> **Q3: Is the choice of an unsupervised framework a necessary outcome for addressing the reconstruction problem of irregularly undersampled transients?**
>
> **Reply:** This is a very insightful question. We believe that the unsupervised framework is not only an effective approach for NLOS imaging from IUT but also a promising paradigm for deep learning in NLOS imaging research.
>
> In recent years, learning-based algorithms have gained wide attention in the NLOS field. Their powerful function fitting and prior learning capabilities allow them to achieve better performance upper bound compared to model-based algorithms.
> However, most of these methods rely on supervised learning with simulated data, which creates a gap between simulated and real-world data, limiting their real-world performance.
> Furthermore, supervised learning requires paired datasets, which makes the model be prone to overfitting to the training domain, leading to poor generalization.
>
> In contrast, unsupervised learning does not rely on paired data, allowing models to generalize better to new and unseen scenes.
> This approach is crucial for NLOS imaging tasks where recovering diverse hidden scenes is necessary.
> We believe the robust generalization capability of the unsupervised paradigm has been demonstrated by our extensive experiments with various real data and irregular patterns.
> Additionally, the unsupervised paradigm reduces the costs and time associated with acquiring labeled datasets, accelerating research progress.
> Therefore, the choice of an unsupervised framework is crucial for effectively addressing NLOS imaging.
> We will also open-source our code to support further research in the NLOS community.
>
> **Q4: Would introducing more irregularly varied masks further enhance the robustness of the network?**
>
> **Reply:** In our experiment, we found that introducing irregular masks led to unstable training and further reduced network performance.
> Due to the light attenuation effect, only a portion of the transient typically contain useful scene information. Therefore, using irregular masks for training often miss a significant part of the information in fully-sampled transients, leading to anomalous irregularly undersampled transient.
> Since our simulated dataset includes objects of various sizes and poses, these anomalies frequently occur during training, making it difficult for the network to converge and reducing its robustness.
>
> **Q5: How the authors constructed different forward propagation operators H based on the varying irregularly relay surfaces.**
>
> **Reply:**
> We constructed different forward operators $H$ using an operator decoupling method, as described in line 149-152.
>
> For the full-sampled relay pattern, the forward operator is denoted as $H_r$, and the corresponding full-sampled transient is $u_r=H_r\rho$, where $\rho$ is the hidden volume.
> Similarly, for a relay surface numbered $g$, the observed transient $u_g$ is given by $u_g=H_g\rho$.
> By replacing through expere $u_g=M_g\odot u_r$, we can obtain expression that $M_g\odot H_r\rho = H_g\rho$.
> So we composition $M_g\odot H_r$ to constructed forward operator $H_g$. In practice, the composition can be efficiently computed through Hadamard product and the fast Fourier transform.

---

> > ### Comment · Reviewer_8wFw · 2024-08-09
> >
> > Thanks to the authors for the detailed response, which effectively addressed my concerns. I'll raise my rating accordingly.
> >
> > In light of the new paradigm for NLOS—specifically, unsupervised learning for NLOS reconstruction—I believe this approach could significantly enhance the generalization capabilities of existing deep learning methods. The application of this approach to reconstruction from irregular transients strongly supports this view.
> >
> > Please update Q1, Q2, and Q4 in the revised manuscript accordingly.

---

> > > ### Author Response · Authors · 2024-08-09
> > > **Official Comment by Authors**
> > >
> > > Thank you for your positive feedback and for raising your rating. We are pleased that our response effectively addressed your concerns and that you recognize the value of our method in applying unsupervised learning to NLOS imaging.
> > >
> > > We will carefully update Q1, Q2, and Q4 in the revised manuscript as per your suggestion. We appreciate your thoughtful review and the time you've taken to help improve our work.

---

### Official Review · Reviewer_cypv · 2024-07-12

**Soundness:** 2
**Presentation:** 1
**Contribution:** 2
**Rating:** 4
**Confidence:** 4

**Summary:**

This paper proposes a non-line-of-sight (NLOS) imaging method for scenarios where transients are irregularly undersampled on the relay surface. The proposed method includes a SURE-based denoising technique to handle noisy transient data, specifically addressing Poisson noise. Additionally, a novel unsupervised learning network called VSRnet is introduced, enabling consistent reconstruction from different irregularly undersampled transients (IUT) of the same 3D albedo volume through a process termed virtual scanning. Extensive comparisons with recent NLOS methods using both simulated and real IUT data demonstrate the superiority of the proposed approach.

**Strengths:**

+ The virtual scanning process makes a contribution by achieving robust reconstruction from incomplete observations in NLOS imaging through unsupervised learning.
+ The SURE-based denoiser, which accounts for Poisson noise, is a contribution as it appears to be universally applicable for denoising transients.

**Weaknesses:**

1. While the SURE-based denoiser may contribute to handling noisy transient data, its originality is limited, and its connection to IUT is unclear.
1. The assumptions regarding the relay surface are not practical. In a setting like Figure 1, the BRDF of the relay surface would vary among the transient samples. However, in the experimental setting, "we extracted signals from the complete transients according to various irregular relay surfaces" (207), which overly simplifies conditions.
1. The justification for zero padding is not clear and it might be unfair for other methods. For instance, instead of treating regions with no samples as zero, using simple interpolation methods like bilinear interpolation might improve the quality of the competing methods.
1. The validation of the proposed virtual scanning is insufficient. While it is claimed that virtual scanning complements the null space (159), this explanation is intuitive rather than theoretically substantiated. There would be no guarantee that "a set of operators $\mathcal{H}$" (170) will fully complement the null space. For example, while it is mentioned that the sampling pattern for virtual scanning is random (138), there is no discussion or experimental validation on the necessary sampling ratio. Although the proposed method conducts the virtual scanning only once for $\rho^{(2)}$, it can be conducted iteratively. However, the iteration might also lead to deviations from observations. This aspect is not adequately discussed.
- Minor comments:
  - The meanings of notations such as $\hat{}$, $\tilde{}$, and various subscripts are not clear enough.
  - Variables like $l$ (86), $G$ (131), and $k\neq g$ (149) are not explained.
  - The process of obtaining $\rho^{(1)}$ and $\rho^{(2)}$ (154) is described in text but not well defined in equations, making it difficult to understand correctly.
  - Referring to the 3D albedo volume $\rho$ as the latent 3D volume (133) seems inappropriate.
  - The citation format might be wrong.

**Questions:**

If there are any misunderstandings in the weaknesses pointed out, please clarify them.

**Limitations:**

The limitations are mentioned only in the supplementary material and not referenced in the main text.

---

> ### Author Rebuttal · Authors · 2024-08-07
>
> We thank the reviewer for valuable comments. We address the main questions as follows.
>
> **Q1: Connection and novelty of our SURE-based denoiser.**
>
> **Reply:** In the irregularly undersampling, the quality and stability of reconstruction could be severely affected by noise, necessitating a robust denoiser to rescue. As shown in Fig. 2 (see experiment setup in the global rebuttal), the PSNR improvement of our method with SURE denoiser over the one without denoiser is up to more than 7 dB at sampling rates between 1.0\% $\sim$ 4\%. With the SURE denoiser, the operating range of sampling rates is broadened to low rate end, which significantly extend the applicability of our method.
>
> Regarding the originality, the SURE denoiser is specifically tailored for NLOS imaging. First, we incorporated the noise model of time-resolved detectors (see Sec. 4.3), addressing the dark noise $b$ in transients, and derived the SURE loss, which is not a straightforward modification of an existing one.
> Second, we proposed a neural network combining partial convolution and instance normalization (IN) layers (see Sec. A).
> Partial convolution contributes to suppress the loss of IUT information during feature propagation through the vanilla convolution and IN layers make network suitable to various distributions of IUTs with different sampling rates.
>
> **Q2: The assumption of relay surfaces.**
>
> **Reply:** Presently, NLOS imaging research community commonly assumes that relay surfaces exhibit uniform diffuse reflectance. However, to our knowledge, none previous methods have taken into account relay surfaces with varying BRDFs. We thank the reviewer poses a challenging task for achieving more practical NLOS imaging and warrants in-depth investigation in future studies.
>
> **Q3: The padding (interpolation) mode for unobserved points.**
>
> **Reply:** We further evaluate two interpolation methods, the bilinear method and nearest-neighbor method, as the pre-processing of reconstruction from IUTs with four baseline methods (LCT, FK, RSD and USM).
> As shown in Figure 4, the performance of interpolators on the final reconstruction is really dependent on the quality of interpolated transient signals, and essentially on the scanning patterns. For scanning patterns with relatively uniform scanning points (small holes on the relay surface, such as Window 3 and 5), the bilinear interpolater can help significantly suppress background reconstruction artifacts. However, scanning patterns with skewed distributed scanning points (large missing areas on the relay surface, such as Window 7), the bilinear interpolater can even degrade the reconstruction quality due to the unreliable interpolated transients in large missing areas.
> For all cases, our method still outperforms than other four baselines.
>
> Since there is not an all-conquering one, we felt that using the simple zero padding is a reasonable comparison setup. Another choice is to adaptively select interpolation methods according to the scanning pattern, e.g. bilinear interpolation for patterns with more uniform sampling and zero padding with skewed sampling. We will discuss more above the padding mode in the final version.
>
> **Q4: Null-space learning of the virtual scanning.**
>
> **Reply:**
> 1. This could be a misunderstanding.
> Our goal is trying to recover the null space components of the reconstruction target (line 159). For this, We provide an intuitive illustration in Fig. 3 of the manuscript and some theoretical explanations in lines 159-171.
> For the special task of NLOS imaging from IUT, some portions of information about the hidden scene may not be captured due to light attenuation. So there is no guarantee that the proposed virtual scanning could complement the entire null space. However, adding various operators for training can effectively improve the generalization of our method. We will discuss more about this in our final version.
>
> 2. The phrase ''randomly sampled from the surface base'' (line 138) indicates that during training, we select one sampling pattern different from $M_g$ from the group of 200 sampling patterns during training to learn the null space components.
>
> 3. Indeed, the sampling rate for each sampling pattern should not be too low for stable and acceptable reconstruction.  We conducted a quantitative experiment, which is detailed in the global rebuttal. Figure 2 illustrates a significant drop in reconstruction quality when the sampling rate falls below 6\%. Notably, all sampling rates for patterns in the surface base exceed this lower bound.
>
> 4. There are two main reasons for the design of virtual scanning:
> First, more iterations in virtual scanning would accumulates errors, causing severe fluctuations in losses and hindering network convergence, which affects final the performance of network. This would also increase required computation.
> Second, in practice, we iterate through each pair of sampling patterns from the surface base to make network learn the null space of the associated operator. This training strategy successfully accelerates network convergence, stabilizes parameter updates, and improves reconstruction quality.
>
> **Q5: Some minor comments.**
>
> **Reply:** The notation $\tilde{}$ refers to the noisy version of transients, and $\hat{}$ refers to their denoised version. $l$ refers the spatial dimension of recovered volume and $G$ is the total number of the operators group. $k\neq g$ means that $k$ is a different number from $g$ and hence their corresponding sampling patterns are also different.
>
> Thank you for pointing out these issues. We will add an additional section to clarify all symbols and variables. We will add more descriptions on the obtaining process of $\rho^{(1)}$ and $\rho^{(2)}$, and updates other presentation issues in the camera-ready version.

---

> > ### Comment · Reviewer_cypv · 2024-08-11
> > **Reply for the rebuttal**
> >
> > The additional experiments presented in the global rebuttal are informative and address some of my concerns.
> >
> > ### Q1: Connection and novelty of the SURE-based denoiser.
> >
> > I appreciate the additional experiment shown in Fig. 2 of the global rebuttal. It clearly demonstrates the effectiveness of the SURE-based denoiser. It shows that a 1.1% sampling rate in the red plot (with the SURE-denoiser) is approximately equivalent to the 100% sampling rate in the blue plot (without the SURE-denoiser) . While using the Poisson distribution for modeling the SPAD sensor is quite common, and I initially considered this work a straightforward extension of existing methods, the effectiveness demonstrated in Fig. 2 raises my rating on it, even though it may seem like an incremental contribution.
> >
> > However, my remaining concern is that the sampling rates are unclear in the reconstruction results (in Figs. 4-7 of the original submission and in Figs. 3 and 4 of the global rebuttal). The visual quality could vary significantly at sampling rates between 1% and 4%, which the authors emphasize. If the authors also show the sampling rates used for each of the results, Fig.2 would be more informative.
> >
> > ### Q2: The assumption of relay surfaces.
> >
> > As other reviewers also raised similar concerns under "Originality/Significance" (WCZG) and as "a concern" (wGeV), the focus on irregular sampling on the relay surface seems far from real-world situations. The focus should be more on non-uniform diffuse reflectance.
> >
> > ### Q3: The padding (interpolation) mode for unobserved points.
> > ### Q4: Null-space learning of the virtual scanning.
> >
> > I appreciate the authors providing the additional experiment shown in Fig. 4 of the rebuttal. However, I would like to see quantitative results for other interpolation methods as well.
> >
> > My main concern with "zero padding" is that it might misinform the existing network that "the space is empty" rather than "the measurement is missing." This seems unfair to the existing methods. Based on the effectiveness of the SURE-denoiser shown in Fig. 2 of the global rebuttal, I still have doubts about the contribution of "virtual scanning," which the authors claim is the main contribution of this paper.
> >
> > Did the authors quantitatively clarify whether the proposed virtual scanning still offers a significant advantage over the combination of "SURE denoiser" and "other interpolation methods" on "LCT, FK, RSD, USM"?

---

> > > ### Author Response · Authors · 2024-08-12
> > > **Official Comment by Authors**
> > >
> > > Thank you very much for your detailed feedback and for recognizing the value of the additional experiments we provided in the rebuttal. We are glad that our response addressed some of your concerns and we appreciate the opportunity to address your remaining concerns in more details.
> > >
> > > **Q1: Connection and novelty of the SURE-based denoiser.**
> > >
> > > We are glad to know that the informative results in Figure 2 of the global rebuttal clarify the effectiveness of the proposed SURE-based denoiser.
> > > It is important to emphasize that the extensive real-world experimental results demonstrate that our method consistently achieves superior reconstruction quality with different sampling rates compared to other algorithms, strongly validating the generalization of our approach to real scenes. The sampling rates for the test cases in Fig. 3 and 4 in the global rebuttal, and Fig. 4$\sim$7 in the manuscript are within the range of 6.25\% $\sim$ 50\%.
> > > We commit to following your suggestion by providing the corresponding sampling rates for the qualitative results in the camera-ready version of the paper.
> > >
> > >
> > > **Q2: The assumption of relay surfaces.**
> > >
> > > We understand your concern regarding the practical relevance of this assumption and appreciate the opportunity to address it.
> > > We agree that exploring NLOS imaging with non-uniform BRDFs is a valuable problem, but it is a parallel challenging problem to our current work, and deserves dedicated efforts from the entire NLOS community.
> > >
> > > Looking back, when NLOS imaging was first introduced, it faced far more limitations than now, including restricted scene conditions, limited relay surfaces, low resolution and reconstruction quality, and slow acquisition and processing times. Over the past decade, improvements (incremental some times) and breakthroughs have been built on the collective efforts of the NLOS research community addressing each subproblem/aspect.
> > > However, it is important to note that NLOS imaging is still primarily a laboratory-scale problem, with many challenges yet to be addressed for practical deployment.
> > > Our work focusing on NLOS imaging from IUT is significant, as it liberates NLOS research from the reliance on large and continuous relay surfaces (such as walls and floors).
> > >
> > > As recognized by reviewers, our work stands out for its well-conducted experiments, thorough ablations, novel analysis, and superior performance compared to existing methods.
> > > Meanwhile, the main technical contributions have been recognized by the reviewers.
> > > Notably, the proposed unsupervised paradigm has been praised for potentially enhancing the generalization capabilities of existing deep learning methods, as noted by Reviewer 8wFw.
> > > In addition, our response has been acknowledged by the reviewers and addressed most concerns from them, including the applicability to more complex irregular relay patterns (Reviewer wGeV, you also mentioned).
> > >
> > > We are confident in our paper and are prepared to open-source our code to enable other NLOS researchers to build on our work and advance practical NLOS applications.
> > >
> > > **Q3: The padding (interpolation) mode for unobserved points.**
> > >
> > > **Q4: Null-space learning of the virtual scanning.**
> > >
> > > Following your suggestions on providing quantitative results for variants of competing methods combined with the SURE denoiser and other two interpolation methods. For  testing, we simulated a test dataset of 1,000 transients by rendering objects (chairs, clocks, guitars, sofas, motorcycles) with random scales and positions. The quantitative results are as follows:
> > > | Method          | LCT (neighbor) | LCT (bilinear) | FK (neighbor) | FK (bilinear) |
> > > |-----------------|----------------|----------------|---------------|---------------|
> > > | PSNR (dB)   | 17.63          | 18.19          | 15.69         | 15.95         |
> > >
> > > | Method          | RSD (neighbor) | RSD (bilinear) | USM (neighbor) | USM (bilinear) | Ours  |
> > > |-----------------|----------------|----------------|----------------|----------------|-------|
> > > | PSNR (dB)   | 15.58          | 15.81          | 14.32          | 14.74          | 20.52 |
> > >
> > > As evident from the quantitative results in the table, our method still offers a significant quantitative improvement over other methods, which confirmed the importance and contribution of virtual scanning.
> > >
> > > Many thanks once again for your valuable feedback.

---

> > > ### Author Response · Authors · 2024-08-13
> > > **Appreciation for the Increased Rating and Continued Feedback**
> > >
> > > Thank you very much for your valuable feedback throughout the author-reviewer discussion, and especially for raising your rating. We greatly appreciate your time and effort in helping us improve our work.
> > >
> > > In our previous responses, we added extensive quantitative and qualitative experiments (e.g., Figures 2 and 4 in the global rebuttal and the table in the comment) to address your concerns, as well as many detailed explanations of our contributions.
> > >
> > > To ensure that we have fully addressed all of your concerns, could you please let us know if there are any remaining aspects of our response that require further clarification or improvement?
> > > If you have any additional feedback or suggestions regarding the revisions we've made, we would be very grateful to receive them and make any necessary adjustments in the camera-ready version. Our goal is to address all of your concerns as thoroughly as possible and to earn your further approval, just as we have with the other reviewers.
> > >
> > > Thank you once again for your review and feedback. We look forward to your further input.

---

### Official Review · Reviewer_WCZG · 2024-07-13

**Soundness:** 3
**Presentation:** 3
**Contribution:** 3
**Rating:** 5
**Confidence:** 2

**Summary:**

The authors attempt the problem of NLOS imaging in the irregularly undersampled transients data case i.e. where the scan pattern on the relay wall isn’t dense or regular. To tackle the problem the authors introduce two main components (both trained unsupervised):

1) A SURE-based denoiser, which denoises the input transients

2) A VSRNet, a “virtual scanning” module, where the estimated albedo is then transformed back into an undersampled transient, and the albedo is estimated again from this sample. VSRNet is trained with an MC loss, enforcing consistency between the two albedo estimates.

The authors justify their choices by arguing about the null space of the light-transport matrix.
The method is benchmarked against a suite of datasets, one of which is also self-collected.

**Strengths:**

Quality: The experiments are well conducted, with great ablations. I especially like that the authors use the SURE denoiser for other methods as well, better justifying the VSRNet. I also like the idea behind VSRNet itself. I think its also great that the authors benchmark the reconstruction speed itself.

Clarity: For the most part I think the paper is well written.

**Weaknesses:**

Originality/Significance: I’m not sure the authors do a great job motivating the problem of irregularly undersampled transients itself. There is a slight motivation in the introduction of scanning through fences etc, but I’m not sure the problem is convincing on its own. I think this also bleeds into a discussion about significance, I’m not sure if the contributions of the paper will be valued enough without a stronger motivation in the introduction/abstract for the problem itself.


Clarity: I appreciate the authors trying to introduce a more theoretically grounded framework, but I’m not sure how important the discussion of the null space actually is. I might have misunderstood some parts, but if I understand correctly the problem is just fundamentally ill-defined, and we could have done without the discussion between lines 100-114.

I think it’s usually good practice to have the method understandable through the figure itself, which is why I would suggest that the authors have a better description for Figure 2. I think it would be good if the authors described the separate components shortly there itself.


Typos:

L61: Citation for Manna et al. not hyperlinked.

**Questions:**

1) Did the authors try to see the effects of different patterns on the reconstruction (not just in training as in the supplement)? I can see a bit in Fig 4 and 6, but are there any limits we can expect i.e. how many scan points do we need, are some patterns worse than others etc?

**Limitations:**

Limitations of the method are not really discussed in the paper, but there is an adequate section in the supplement. Societal impact N/A.

---

> ### Author Rebuttal · Authors · 2024-08-07
>
> We are highly encouraged by the positive recommendation and comments from the reviewer on our experiment, method and presentation. Furthermore, the raised questions are both central and valuable. We address the main questions as follows.
>
> **Q1: The significance of NLOS imaging from irregularly undersampled transients.**
>
> **Reply:** The problem of irregularly undersampled transients in NLOS imaging is critical for practical applications.
> In real-world scenarios, relay surfaces are often discrete and irregularly shaped, such as fences, window shutter and window frame.
> Most current algorithms rely on dense and continuous transients, limiting their practical application.
> Developing algorithms for IUTs is essential to extend NLOS imaging to these common environments.
> Our method addresses this by efficiently using sparse transient information, significantly broadening the applicability of NLOS imaging.
>
>
> **Q2: The importance of the discussion of the null space.**
>
> **Reply:** The discussion in lines 100-114 is crucial for understanding the foundation of our proposed **unsupervised learning** algorithm, as it explains why our virtual scanning strategy is necessary and effective.
>
> Specifically, our method is inspired by the range-null decomposition [1] and null space learning [2][3][4].
> We observed that using only measurement consistency loss (MC loss) is insufficient for high-quality reconstruction. This is because only using MC loss for training cannot recover the null-space component of the target. To this end, we propose the virtual scanning strategy to recover the missing null-space component for high quality reconstructions, as shown in Figure 3(c) of the manuscript.
>
> It establishes the theoretical foundation for our approach, elucidating the rationale behind our method and emphasizing how we address the limitations of prior techniques. Without this theoretical context, readers might question why the pipeline performs exceptionally well.
>
> **Q3: The effects of different patterns on the reconstruction and
> some limits we can expect.**
>
> **Reply:** Thank you for this very insightful observation and suggestion.
> In Figure 1(b) (see global rebuttal), we illustrate how the missing pattern affect the reconstruction quality. In general, a relay surface with a large missing area often results in poor reconstruction quality due to the loss of information. Even with the same sampling rate, more uniform scanning point distribution yields better results (e.g., the fifth column vs. the second).
>
> As shown in Figure 1(a) (global rebuttal), light attenuation term $1/r^4$ causes that a histogram ($1\times1\times T$) captured from scanning point $p$ often only contain information from hidden areas close to the scanning point $p$, such as the green oval area shown.
> This leads to poor reconstruction if large parts of the relay surface are missing.
> However, the acceptable size of the missing area for reconstruction is hard to derive theoretically. As it depends on multiple factors such as the relative depth between the scene and the relay surface, the scene's reflectivity, normals, detector efficiency, and laser power.
>
> Nevertheless, we tried to obtain a general trend through quantitative experiments.
> We simulated a dataset of 1,000 transients by rendering objects (chairs, clocks, guitars, sofas, motorcycles) with random scaling and positioning.
> The relay surface consisted of random points with sampling rates ranging from 0.1\% to 100\%, forming 30 groups.
> As shown by the red curve in Figure 2, the PSNR of the reconstructed intensity map starts to decline sharply below the sampling rate of 6\%. Therefore, we recommend that the sampling rates for patterns should larger than 6\%. We will add these discussion in the final version.
>
> **Q4: Presentations about Figure 2 and others.**
>
> **Reply:**
> Thanks for your suggestions on the presentation. We will include additional descriptions of the components in Figure 2, such as the networks $F_\theta$ and $F_\phi$, in the camera-ready version.
> Additionally, we will repair the hyperlinks in the citations and move the limitations section to the main text.
>
> [1] Schwab, Johannes, Stephan Antholzer, and Markus Haltmeier. "Deep null space learning for inverse problems: convergence analysis and rates." Inverse Problems 35.2 (2019): 025008.
>
> [2] Sønderby, Casper Kaae, et al. "Amortised MAP Inference for Image Super-resolution." International Conference on Learning Representations. 2022.
>
> [3] Wang, Yinhuai, et al. "Gan prior based null space learning for consistent super-resolution." Proceedings of the AAAI Conference on Artificial Intelligence. Vol. 37. No. 3. 2023.
>
> [4] Wang, Yinhuai, Jiwen Yu, and Jian Zhang. "Zero-Shot Image Restoration Using Denoising Diffusion null space Model." The Eleventh International Conference on Learning Representations.

---

> ### Comment · Reviewer_WCZG · 2024-08-11
>
> Thank you very much to the authors for their responses.
>
>
> I'm not sure my comments on the motivation for the paper have been adequately addressed. But I'm not sure this can be addressed.
> Nonetheless, I appreciate the general rebuttal, especially Figure 1, I think it's quite interesting to see the effects of the reconstructions with the patterns the authors show, and I would suggest the authors add this discussion to the paper.
>
>
> I'm not yet sure I will increase my score, I will be following the author's discussions with the other reviewers to set my final score, thanks!

---

> > ### Author Response · Authors · 2024-08-12
> > **Official Comment by Authors**
> >
> > Thank you very much for your continued engagement and constructive feedback.
> > We can feel your strong sense of responsibility and rigorous academic standards from your comments.
> >
> > We're pleased that you found our general rebuttal and the insights from Figure 1 valuable. Per your suggestion, we will include a detailed discussion of these results in the camera-ready version of the paper, highlighting the impact of irregular sampling.
> >
> > Regarding your concerns about the motivation of our work, we understand that our initial explanation may not have fully addressed your questions.
> > Looking back, when NLOS imaging was first introduced, it faced far more limitations than now, including restricted scene conditions, limited relay surfaces, low resolution and reconstruction quality, and slow acquisition and processing times. Over the past decade, improvements (incremental some times) and breakthroughs have been built on the collective efforts of the NLOS research community addressing each subproblem/aspect.
> > However, it is important to note that NLOS imaging is still primarily a laboratory-scale problem, with many challenges yet to be addressed for practical deployment.
> > Our work focusing on NLOS imaging from IUT is significant, as it liberates NLOS research from the reliance on large and continuous relay surfaces (such as walls and floors).
> >
> > As recognized by reviewers, our work stands out for its well-conducted experiments, thorough ablations, novel analysis, and superior performance compared to existing methods.
> > Meanwhile, the main technical contributions have been recognized by the reviewers.
> > Notably, the proposed unsupervised paradigm has been praised for potentially enhancing the generalization capabilities of existing deep learning methods, as noted by Reviewer 8wFw.
> > We will open-source our code to enable other NLOS researchers to build on our work and advance practical NLOS applications.
> >
> > We are committed to improving the paper in our camera-ready version based on your and other reviewers' feedback, and we welcome any additional suggestions you may have.
> > Thanks once again for your valuable input.

---

### Author Rebuttal · Authors · 2024-08-07

**Figure 1:** As suggested by reviewers WCZG, 8wFw and wGeV who concern about the effects of different relay patterns, we address this issue by analysing the NLOS imaging model and corresponding experiments.

Figure 1(a) shows the top view of a typical confocal imaging system.
Different colored lines indicate light rays illuminating to different positions in the hidden scene, and transparency of them represents light intensity.  The light attenuation term $1/r^4$ causes that a histogram ($1\times1\times T$) captured from scanning point $p$ often only contain information from hidden areas close to the scanning point $p$, such as the green oval area shown. Photons from farther areas attenuate due to diffuse reflection term $1/r^4$, making them almost undetectable.

In Figure 1(b), we illustrate how the missing pattern affect the reconstruction quality. In general, a relay surface with a large missing area often results in poor reconstruction quality due to the loss of information. Even with the same sampling rate, more uniform scanning point distribution yields better results (e.g., the fifth column vs. the second).

Similarly, using irregular relay patterns for training can miss a significant part of the information in fully-sampled transients, generating anomalous IUT.
Since the simulated dataset is rendered from objects of various sizes and poses, these anomalous IUTs frequently occur during training, making the network difficult to converge and ultimately reducing its robustness.

**Figure 2:**
As suggested by reviewers WCZG, cypv and wGeV who concern about necessary sampling ratio, we address this issue by adding quantitative experiments on different sampling ratios.

Specially, We simulated a dataset of 1,000 transients by rendering objects (chairs, clocks, guitars, sofas, motorcycles) with random scaling and positioning.
The relay surfaces for testing are consisted of random points with sampling rates ranging from 0.1\% to 100\%, forming 30 groups.

As shown by the red curve in Figure 2, the PSNR of the reconstructed intensity map starts to decline sharply below the sampling rate of 6\%. Therefore, we recommend that the sampling rates for patterns should larger than 6\%.

In addition, the PSNR improvement of our method with SURE denoiser over the one without denoiser is up to more than 7 dB at sampling rates between 1.0\% $\sim$ 4\%. With the SURE denoiser, the operating range of sampling rates is broadened to low rate end, which significantly extend the applicability of our method.

**Figure 3:**
As suggested by reviewer wGeV who concerns about our method's generalization to irregular relay surfaces that is not composed of simple repetitive or symmetrical shapes, we address this issue by adding qualitative experiments.

We tested our method on publicly available real datasets with more irregular relay surface. These irregular relay surfaces do not contain simple repetitive or symmetrical shapes, such as the signage of NIPS 2024, window papercut with various shapes and even random graffiti. As shown in Figure 3, our method still achieves the best quality for various sampling patterns. In addition, as shown in Fig. 5 and 6 in the manuscript, the relay surface ''random points'' and ''regular 32$\times$32'' are more different from our training patterns, but our method can also handle them well. This is because our method's generalization is guaranteed by the theoretically grounded framework in learning beyond the range space.

**Figure 4:**
As suggested by reviewer cypv who concerns about the pre-processing methods of the baselines, we address this issue by adding comparison experiments.

We further evaluate two the interpolation methods, the bilinear method and nearest-neighbor method,as the pre-processing of reconstruction from IUTs with four baseline algorithms (LCT, FK, RSD and USM).
As shown in Figure 4, the performance of interpolators on the final reconstruction is really dependent on the quality of interpolated transient signals, and essentially on the scanning patterns. For scanning patterns with relatively uniform scanning points (small holes on the relay surface, such as Window 3 and 5), the higher-order bilinear interpolater can help significantly suppress background reconstruction artifacts. However, scanning patterns with skewed distributed scanning points (large missing areas on the relay surface, such as Window 7), the higher-order bilinear interpolater can even degrade the reconstruction quality due to the unreliable interpolated transients in large missing areas.
For all cases, our method still outperforms than other four baselines.

---

### Decision · Program_Chairs · 2024-09-25

**Decision:**

Accept (poster)

**Comment:**

The paper received positive reviews overall that became stronger after the authors’ rebuttal. Reviewers highlighted the extensive evaluations (wGeV, 8wFw, WCZG), the novel analysis for unsupervised methods (8wFw), and the derivation of SURE loss (8wFw, WCZG). Although there were initial concerns regarding irregularity of scanning patterns, unclear presentations, missing experiments, and motivations, these were effectively addressed in the rebuttal, leading to stronger overall reviews.

After carefully reviewing the paper, the reviews, and the rebuttal, the AC agrees with the reviewers’ positive consensus, and hence recommends acceptance of the paper.

For the camera-ready revision, the authors should ensure to incorporate all discussions from the rebuttal into the main paper. Specific changes to be implemented are:

**1. Additional experiments**: Include the quantitative evaluation results for VSRnet and SURE-denoiser (8wFw), and the experiments and discussion for the effects of different relay patterns and more irregularly varied masks (WCZG, 8wFw, wGeV)

**2. Improved presentation**: Clarify all symbols and schematics (8wFw, cypv), update citation format (cypv) and hyperlinks (WCZG), and improve figure 2 (WCZG)